# When resampling/reweighting improves feature learning in imbalanced classification?: A toy-model study

## Abstract

A toy model of binary classification is studied with the aim of clarifying the class-wise resampling/reweighting effect on the feature learning performance under the presence of class imbalance. In the analysis, a high-dimensional limit of the feature is taken while keeping the dataset size ratio against the feature dimension finite and the non-rigorous replica method from statistical mechanics is employed. The result shows that there exists a case in which the no resampling/reweighting situation gives the best feature learning performance irrespectively of the choice of losses or classifiers, supporting recent findings in Cao et al. (2019); Kang et al. (2019). It is also revealed that the key of the result is the symmetry of the loss and the problem setting. Inspired by this, we propose a further simplified model exhibiting the same property for the multiclass setting. These clarify when the class-wise resampling/reweighting becomes effective in imbalanced classification.

## 1 Introduction

Real-world datasets for classification occasionally exhibit strong class imbalance with a long-tailed class distribution (Van Horn et al., 2018; iNa; Liu et al., 2019). Classifiers applied to such datasets tend to perform poorly for minority classes, which poses a major challenge in areas such as visual recognition. Although several methods to mitigate class imbalance have been proposed so far (Chawla et al., 2002; He & Garcia, 2009; Wallace et al., 2011), recent advances of deep learning have shed new light on this issue, resulting in numerous studies from the perspective of applying those approaches to classifiers based on deep neural networks (DNNs) (Liu et al., 2019; Huang et al., 2016; Wang et al., 2017; Cui et al., 2018; Khan et al., 2019; Cui et al., 2019; Cao et al., 2019; Kang et al., 2019; Jamal et al., 2020; Tan et al., 2020; Menon et al., 2020; Kini et al., 2021).

Among those approaches proposed so far, we focus on two simple strategies, reweighting and resampling, which are commonly employed to mitigate class imbalance. The resampling strategy tries to balance the samples in the dataset by oversampling the minority classes and/or undersampling the majority classes, while the reweighting strategy puts an additional weight to each term of the loss in order to counterweight the class imbalance. The effectiveness of these strategies has been empirically verified in a wide range of studies (Cui et al., 2019; Cao et al., 2019; Kang et al., 2019; Jamal et al., 2020; Chawla et al., 2002; He & Garcia, 2009). In spite of these pieces of work, transparent description or understanding about when they are useful or not would still be imcomplete. In particular, how class imbalance may affect the quality of feature learning would be an important problem in the context of representation learning in DNNs, but a thorough understanding of this issue is still missing.

Recently, Kang et al. (2019) reported an interesting observation that feature learning becomes better if no resampling is applied. More specifically, on the basis of their extensive experiment on visual recognition tasks using DNNs, they reported that the best classification performance was achieved when the whole network was first trained without any resampling and then only the last output layer (final classifier) was retrained with class-balanced resampling. This observation can be interpreted as follows: one can learn the best feature representation in the initial training phase if one does not use resampling, and the good classification performance achieved by the retrained network is ascribed to exploitation of the good feature representation

acquired in the initial training. A similar behavior was also reported in Cao et al. (2019). One can therefore expect that Kang et al.'s observation would provide a useful generic insight into efficiency of resampling with regard to feature learning.

In this paper, we provide a theoretical analysis on a toy model to examine the effect of resampling and reweighting, especially aiming to clarify under what conditions the observation by Kang et al. holds. In our toy model, we treat a binary classification problem, in which the sample-generation process is assumed to be stochastic. More specifically, samples are independent and identically-distributed (i.i.d.) from a probability distribution on $\mathbb{R}^N$, with parameters controlling the class imbalance and the variances. The two class centers are assumed to be located at $\pm \boldsymbol{w}_0/\sqrt{N} \in \mathbb{R}^N$. These constitute a standard setting for theoretical treatments of binary classification (Barkai & Sompolinsky, 1994).

In the above setting, the performance of feature learning can be quantified as the accuracy of estimating $\boldsymbol{w}_0$, since it represents the most discriminative direction of the two classes under isotropic class-conditional sample distributions. Our analysis, which considers the asymptotic $N \to \infty$, reveals that, under the condition that the variances of samples of the two classes are equal, the accuracy of estimating $\boldsymbol{w}_0$ is maximized when one does not employ resampling/reweighting at all, irrespectively of the degree of the class imbalance. This finding provides an analytical support for Kang et al.'s observation. We would also like to emphasize that this finding remains valid with a rather wide range of classifiers and losses, which can be understood from the symmetry in formulae derived in our analysis. Although the equal-variance condition for sample distributions might be seem somewhat artificial, it may be achieved in the feature representation at the last hidden layer of a classification DNN as a result of DNN training, and some recent studies partially support this (Papyan et al., 2020; Fang et al., 2021).

Our results are derived via statistical-mechanical techniques (Barkai & Sompolinsky, 1994; Barkai et al., 1993; Biehl & Mietzner, 1993; Watkin & Nadal, 1994; Lootens & van den Broeck, 1995; Tanaka, 2013), which are applicable in the limit $N \to \infty$. Especially, the replica method (Nishimori, 2001; Dotsenko, 2005; Mezard & Montanari, 2009) plays a key role in computing the quantities of interest. Although the replica method is mathematically non-rigorous, the results derived in this paper are conjectured to be correct, which is supported by an excellent agreement with numerical experiment, as well as by an accumulation of many model analyses for long decades in which the replica method is eventually shown to give the exact results (Montanari & Tse, 2006; Talagrand, 2003; 2011a;b).

The remainder of this paper is organized as follows. In the next section, the problem setup and the formulation are explained. In sec. 3, the analysis details using the replica method are explained. The derived formulae using the replica method are utilized to systematically examine behaviors of the quantities of interest. The result shows that there exists a case in which the absence of resampling/reweighing gives the best feature learning performance irrespectively of the choice of losses or classifiers, yielding a theoretical support for Kang et al.'s observation. On the basis of this theoretical result, we also provide a further simplified model for multiclass classification, for which the same consequence about the resampling/reweighting holds. In sec. 4, the numerical experiments are conducted to verify the replica result. The last section concludes the paper.

## 2 Problem setup and formulation

### 2.1 Data-generation process

Let us consider a classification problem with two classes labeled by $y = \pm 1$ whose distribution is

$$P_Y(y) = \sum_{y'=\pm 1} r_{y'} \delta_{y,y'}, \tag{1}$$

where $\delta_{a,b}$ denotes the Kronecker delta and where $r_{\pm 1} \in [0,1]$ with $r_{+1} + r_{-1} = 1$ control the degree of class imbalance: $r_{+1} = r_{-1} = 1/2$ corresponds to the balanced case. The input space is assumed to be $\mathbb{R}^N$ and the input vector $\boldsymbol{x} \in \mathbb{R}^N$ is assumed to be generated from the following model given a label $y \in \{-1, 1\}$:

$$\boldsymbol{x} = y \frac{\boldsymbol{w}_0}{\sqrt{N}} + \boldsymbol{\xi}(y), \tag{2}$$

where $\pm\boldsymbol{w}_0/\sqrt{N} \in \mathbb{R}^N$ represent the class centers corresponding to the two classes $y = \pm 1$. We impose the normalization condition $\|\boldsymbol{w}_0\|^2/N = 1$. In Eq. (2), $\boldsymbol{\xi}(y)$ is assumed to be a random i.i.d. vector obeying the zero-mean Gaussian distribution

$$P_{\Xi|Y}\left(\boldsymbol{\xi} \mid \sigma_y^2\right) = \left(2\pi\sigma_y^2\right)^{-N/2} e^{-\frac{1}{2\sigma_y^2}\|\boldsymbol{\xi}\|^2}, \tag{3}$$

where the label-$y$ dependence appears only through the variance $\sigma_y^2$ which is finite for both $y = \pm 1$. The variance $\sigma_y^2$ in a sense expresses the cluster size of the class $y$ in the input space. Let $D^M = \{(\boldsymbol{x}_\mu, y_\mu)\}_{\mu=1}^M$ be a dataset of $M$ i.i.d. datapoints following the above data-generation process:

$$P(D^M \mid \boldsymbol{w}_0) = \prod_{\mu=1}^M r_{y_\mu} P_{\Xi|Y}\left(\boldsymbol{x}_\mu - y_\mu \frac{\boldsymbol{w}_0}{\sqrt{N}} \,\Big|\, \sigma_{y_\mu}^2\right). \tag{4}$$

Although we derive our results on the basis of the Gaussian assumption (3), we expect that the same results hold for a much wider class of distributions with $\sigma_y^2 < \infty$ thanks to the universality appearing through the central limit theorem in the limit $N \to \infty$.

## 2.2 Classifier and loss

A generic classifier can be formulated as first mapping the input $\boldsymbol{x}$ onto a one-dimensional feature $z = f(\boldsymbol{x}) \in \mathbb{R}$ via a feature mapping $f$, and then, on the basis of the feature $z = f(\boldsymbol{x})$, producing a soft decision $P(y \mid \boldsymbol{x})$ which is an estimate of the conditional distribution of the class label $y$ for the input $\boldsymbol{x}$. In our formulation, we consider the simplest case where a linear feature mapping $f(\boldsymbol{x}) = \boldsymbol{w}^\top \boldsymbol{x}/\sqrt{N}$ is to be used, with $\boldsymbol{w} \in \mathbb{R}^N$ the parameter of the feature mapping $f$. As the soft-decision classifier given the one-dimensional feature $z = f(\boldsymbol{x})$, which we call the feature-based classifier, we assume $P(y \mid \boldsymbol{x}) = \mathcal{M}(y(f(\boldsymbol{x}) + b))$ with a function $\mathcal{M} : \mathbb{R} \to [0,1]$. The bias term $b$ is the parameter of the feature-based classifier. We may also write $P(y \mid \boldsymbol{x}; \boldsymbol{w}, b) = \mathcal{M}(y(\boldsymbol{w}^\top \boldsymbol{x}/\sqrt{N} + b))$ in order to make explicit the dependence of the classifier on the parameters $\boldsymbol{w}$ and $b$. By an appropriate choice of the function $\mathcal{M}$, this formulation covers several standard classifier models, such as a perceptron and a logit function:

$$\mathcal{M}_{\mathrm{pe}}(h) \coloneqq \delta_{\mathrm{sgn}(h),1}, \tag{5}$$

$$\mathcal{M}_{\mathrm{lo}}(h) \coloneqq \frac{e^h}{2\cosh(h)}. \tag{6}$$

We say that a feature-based classifier using $\mathcal{M}(h)$ is symmetric if the function $\mathcal{M}$ satisfies $\mathcal{M}(h) + \mathcal{M}(-h) = 1$ for all $h \in \mathbb{R}$.

The weights $\boldsymbol{w}$ are usually determined by minimizing an empirical loss, which is a (sometimes weighted) sum of loss values over all the datapoints. We let $\mathcal{L}(\boldsymbol{w}; \boldsymbol{x}, y, \mathcal{M})$ denote the loss of the weights $\boldsymbol{w}$ given the datapoint $(\boldsymbol{x}, y)$ and the model $\mathcal{M}$. According to the above assumptions of the linear feature mapping and on the model, generic loss function can be written as a function $\ell(h, y)$ of two arguments $h = \boldsymbol{w}^\top \boldsymbol{x}/\sqrt{N} + b$ and $y$, in such a way that the relation $\mathcal{L}(\boldsymbol{w}; \boldsymbol{x}, y, \mathcal{M}) = \ell(\boldsymbol{w}^\top \boldsymbol{x}/\sqrt{N} + b, y)$ holds. A common loss function in the recent practice is cross entropy (CE) and the corresponding loss function takes the form

$$\ell_{\mathrm{CE}}(\boldsymbol{w}^\top \boldsymbol{x}/\sqrt{N} + b, y) = -\log \mathcal{M}(y(\boldsymbol{w}^\top \boldsymbol{x}/\sqrt{N} + b)). \tag{7}$$

As $\ell_{\mathrm{CE}}(h, y)$ depends on $h$ and $y$ only through their product $yh$, it has the symmetry $\ell_{\mathrm{CE}}(-h, -y) = \ell_{\mathrm{CE}}(h, y)$; it is also the case with many other standard losses which are represented as functions of $yh$, such as zero-one, exponential, smoothed or non-smoothed hinge losses. Hereafter $\ell$ is used to denote a generic loss having this symmetry. To express a specific model and loss, we use an appropriate subscript: for example, if we use the logit model and the CE, the resultant loss will be denoted as $\ell_{\mathrm{CElo}}$. Furthermore, if $\ell$ is used with a single argument, we assume that it expresses the one for the positive label: $\ell(h) = \ell(h, +1)$. This is a convenient shorthand notation when we work with the above symmetry.

The empirical loss considered in this paper has class-wise reweighing factors $s_y \in [0, 1]$, which are assumed to satisfy the condition $s_{+1} + s_{-1} = 1$. Given a loss $\ell$ and a dataset $D^M = \{(\boldsymbol{x}_\mu, y_\mu)\}_{\mu=1}^M$, the empirical

reweighted loss is thus written by

$$\mathcal{H}(\boldsymbol{w} \mid D^M; b, s) = \sum_{\mu=1}^{M} s_{y_\mu} \ell(\boldsymbol{w}^\top \boldsymbol{x}_\mu / \sqrt{N} + b, y_\mu). \tag{8}$$

This class-wise reweighting of the loss is intended to mitigate possible undesirable effects of the class imbalance arising when $r_{\pm 1} \neq 1/2$. It can be considered as a special case of the instance-wise weighting which has been investigated, e.g., in He & Garcia (2009). We would also like to mention that resampling offers yet another alternative for the purpose of mitigation of class imbalance. Resampling, however, has an effect of changing the instance-wise weights in the empirical loss, and hence the average effect of the class-wise resampling can be incorporated into this reweighting factor. We analyze the property of the above loss and its minimizer $\hat{\boldsymbol{w}} = \underset{\boldsymbol{w}:\|\boldsymbol{w}\|^2=N}{\arg\min} \mathcal{H}(\boldsymbol{w} \mid D^M; b, s)$ under the constraint $\|\boldsymbol{w}\|^2 = N$, especially focusing on the overlap between $\hat{\boldsymbol{w}}$ and $\boldsymbol{w}_0$ as it quantifies the quality of feature learning. Although estimating the bias $b$ may also be performed via minimization of $\mathcal{H}$ with respect to $b$, the analytical framework explained in the next section allows us to do it in a more flexible manner and we leave $b$ as a tunable parameter.

Before proceeding, we have a noteworthy remark about the Bayesian inference. Suppose that we know the data-generation process but do not know the specific values of either $\boldsymbol{w}_0$ or $r_{\pm 1}$. We thus introduce $\boldsymbol{w}$ as a variable to estimate $\boldsymbol{w}_0$ and $r_{\pm 1}$ as hyperparameters playing the role of class-wise reweighting factor. With an appropriate prior $P(\boldsymbol{w})$, the posterior distribution of $\boldsymbol{w}$ given the dataset $D^M$ becomes

$$
\begin{aligned}
P(\boldsymbol{w} \mid D^M) &= \frac{P(\boldsymbol{w}) \prod_{\mu=1}^{M} r_{y_\mu} P_{\boldsymbol{\Xi}|Y}\left(\boldsymbol{x}_\mu - y_\mu \frac{\boldsymbol{w}}{\sqrt{N}} \mid \sigma_{y_\mu}^2\right)}{\int d\boldsymbol{w} P(\boldsymbol{w}) \prod_{\mu=1}^{M} r_{y_\mu} P_{\boldsymbol{\Xi}|Y}\left(\boldsymbol{x}_\mu - y_\mu \frac{\boldsymbol{w}}{\sqrt{N}} \mid \sigma_{y_\mu}^2\right)} \\
&= \frac{P(\boldsymbol{w}) \prod_{\mu=1}^{M} P_{\boldsymbol{\Xi}|Y}\left(\boldsymbol{x}_\mu - y_\mu \frac{\boldsymbol{w}}{\sqrt{N}} \mid \sigma_{y_\mu}^2\right)}{\int d\boldsymbol{w} P(\boldsymbol{w}) \prod_{\mu=1}^{M} P_{\boldsymbol{\Xi}|Y}\left(\boldsymbol{x}_\mu - y_\mu \frac{\boldsymbol{w}}{\sqrt{N}} \mid \sigma_{y_\mu}^2\right)}.
\end{aligned} \tag{9}
$$

Hence the posterior distribution does not depend on $r_{\pm 1}$, meaning that the reweighting via adjusting $r_{\pm 1}$ has no effect on estimation of $\boldsymbol{w}_0$. If we treat $r_{\pm 1}$ as variables with a prior $P(\{r_{\pm 1}\})$, the result is seemingly different but the marginal posterior $P(\boldsymbol{w} \mid D^M) = \int dr\, P(\boldsymbol{w}, r \mid D^M)$ is still independent of the choice of $P(\{r_{\pm 1}\})$ as long as the priors of $r_{\pm 1}$ and $\boldsymbol{w}$ are independent. Hence, to study the effect of reweighting/resampling on the feature learning, the Bayesian inference framework based on the true data-generation process is inappropriate.

### 2.3  Statistical mechanical formulation

To investigate the estimator $\hat{\boldsymbol{w}} = \underset{\boldsymbol{w}:\|\boldsymbol{w}\|^2=N}{\arg\min} \mathcal{H}(\boldsymbol{w} \mid D^M; b, s)$, it is convenient to introduce the following distribution:

$$P_\beta(\boldsymbol{w} \mid D^M; b, s) := \frac{1}{Z} \delta(N - \|\boldsymbol{w}\|^2) e^{-\beta \mathcal{H}(\boldsymbol{w}|D^M;b,s)}, \tag{10}$$

$$Z = Z(D^M; b, s) := \int d\boldsymbol{w}\, \delta(N - \|\boldsymbol{w}\|^2) e^{-\beta \mathcal{H}(\boldsymbol{w}|D^M;b,s)}, \tag{11}$$

where $\delta(\cdot)$ denotes the Dirac measure, and where $\beta \geq 0$ is the inverse temperature parameter. In the limit $\beta \to \infty$, the distribution $P_\beta$ concentrates on the set of minimizers of $\mathcal{H}(\boldsymbol{w} \mid D^M; b, s)$, and hence any properties of the estimator $\hat{\boldsymbol{w}}$ can be computed from the average over the distribution in the limit. Further, $\phi = -(\beta N)^{-1} \log Z$ plays the role of the cumulant generating function of $\boldsymbol{w}$ and converges to the per-variable average loss in the limit $\beta \to \infty$, that is, $\lim_{\beta \to \infty} \phi = u := \min_{\boldsymbol{w}:\|\boldsymbol{w}\|^2=N} \mathcal{H}(\boldsymbol{w} \mid D^M; b, s)/N$. This means that $\phi$ contains all the necessary information for our purpose and hereafter we concentrate on computing it. According to the physics terminology, in the following we call $P_\beta$ the Boltzmann distribution, $T = 1/\beta$ the temperature, $Z$ the partition function, $\phi$ the free energy, and $u$ the energy. The average over the Boltzmann

distribution is denoted by the angular brackets as

$$\langle(\cdots)\rangle = \int d\boldsymbol{w} P_\beta(\boldsymbol{w} \mid D^M; b, s)(\cdots) = \mathrm{Tr}_{\boldsymbol{w}} \frac{e^{-\beta\mathcal{H}(\boldsymbol{w}|D^M;b,s)}}{Z}(\cdots), \tag{12}$$

where the symbol $\mathrm{Tr}_{\boldsymbol{w}} = \int d\boldsymbol{w}\, \delta(N - \|\boldsymbol{w}\|^2)$ is introduced for notational simplicity of our development later.

A problem arises in the computation of the free energy $\phi$: it depends on the random variable $D^M$ and hence its direct assessment is difficult. However, $\phi$ is expected to exhibit what is called the self-averaging property, implying that it converges to its expectation value over $D^M$ in the limit $N \to \infty$. Hence we may instead compute $[\phi]_{D^M}$, where the square brackets express the average over the data-generation process:

$$[(\cdots)]_{D^M} = \left(\prod_{\mu=1}^M \sum_{y_\mu=\pm 1} \int d\boldsymbol{x}_\mu\right) P(D^M \mid \boldsymbol{w}_0)(\cdots), \tag{13}$$

where $P(D^M \mid \boldsymbol{w}_0)$ is given in (4). Unfortunately, the evaluation of $[\phi]_{D^M} = -(\beta N)^{-1}[\log Z]_{D^M}$ is still difficult. The replica method is a great aid in such a situation, via making use of the following identity:

$$[\log Z]_{D^M} = \lim_{n \to 0} \frac{1}{n} \log [Z^n]_{D^M}. \tag{14}$$

In addition to this identity, we assume that $n$ is a positive integer. This assumption enables us to explicitly compute the average $[(\cdots)]_{D^M}$ on the right-hand side of Eq. (14). After computing this average, we take the limit $n \to 0$ by relying on an expression of the average that is analytically continuable from $\mathbb{N}$ to $\mathbb{R}$, under what is called the replica symmetric (RS) ansatz. The details are in the next section.

## 3 Theoretical Analysis

### 3.1 Overview

In this section we derive a compact formula for the free energy by using the replica method under the RS ansatz. The formula is characterized by a few quantities which are called *order parameters*, and the order parameters obey a set of equations called *equations of state (EOS)*, according to the physics terminology. The order parameters' dependence on the parameters, especially on the reweighting factor $s_{\pm 1}$, is our special focus in the paper and is systematically studied on the basis of the EOS. A further simplified model inspired from the replica results will also be introduced later in this section to discuss the case with more than two classes.

For notational simplicity, we use the shorthand notation $s_+(s_-)$ to denote $s_{+1}(s_{-1})$ hereafter. The same shorthand rule applies to $\sigma_y$ and $r_y$ as well.

### 3.2 Derivation of free energy and EOS under RS ansatz

The computation starts from evaluating $[Z^n]_{D^M}$. If $n \in \mathbb{N}$, we have

$$[Z^n]_{D^M} = \left[\mathrm{Tr}_{\{\boldsymbol{w}_a\}_{a=1}^n} e^{-\beta \sum_{\mu=1}^M \sum_{a=1}^n s_{y_\mu}\ell(\boldsymbol{w}_a^\top \boldsymbol{x}_\mu/\sqrt{N}+b, y_\mu)}\right]_{D^M}, \tag{15}$$

where $\mathrm{Tr}_{\{\boldsymbol{w}_a\}_{a=1}^n} = \prod_{a=1}^n \mathrm{Tr}_{\boldsymbol{w}_a}$. The average over the dataset yields

$$\left[\mathrm{Tr}_{\{\boldsymbol{w}_a\}_{a=1}^n} e^{-\beta \sum_{\mu=1}^M \sum_{a=1}^n s_{y_\mu}\ell(\boldsymbol{w}_a^\top \boldsymbol{x}_\mu/\sqrt{N}+b, y_\mu)}\right]_{D^M} = \mathrm{Tr}_{\{\boldsymbol{w}_a\}_{a=1}^n} L^M, \tag{16}$$

where

$$L := \sum_{y=\pm 1} r_y \int d\boldsymbol{x}\, P_{\boldsymbol{\Xi}|Y}\left(\boldsymbol{x} - y\frac{\boldsymbol{w}_0}{\sqrt{N}} \,\Big|\, \sigma_y^2\right) e^{-\beta \sum_{a=1}^n s_y \ell(\boldsymbol{w}_a^\top \boldsymbol{x}/\sqrt{N}+b, y)}. \tag{17}$$

The integral over $\boldsymbol{x}$ is cumbersome. However, the integrand depends on $\boldsymbol{x}$ only through the quantity $u_a = \boldsymbol{w}_a^\top \boldsymbol{x}/\sqrt{N}$. Conditional on $y$, this quantity $u_a$ obeys a Gaussian, thanks to the assumption (3) in the present setup or thanks to the central limit theorem in the large-$N$ limit in a more generic case. Denoting the average over $\boldsymbol{\xi}$ given $y$ as $\mathbb{E}\left[\cdot \mid y\right]$, the conditional mean of $u_a$ given $y$ is

$$\mathbb{E}\left[u_a \mid y\right] = \frac{\boldsymbol{w}_a^\top \mathbb{E}\left[\boldsymbol{x} \mid y\right]}{\sqrt{N}} = y m_a, \tag{18}$$

where we let $m_a := \boldsymbol{w}_a^\top \boldsymbol{w}_0/N$. The conditional covariance of $u_a$ and $u_b$ given $y$ is

$$\mathbb{E}\left[u_a u_b \mid y\right] - \mathbb{E}\left[u_a \mid y\right]\mathbb{E}\left[u_b \mid y\right] = Q_{ab}\sigma_y^2, \tag{19}$$

where we let $Q_{ab} := \boldsymbol{w}_a^\top \boldsymbol{w}_b/N$. To proceed further, we assume the RS as follows:

$$m_a = m, \quad Q_{ab} = Q\delta_{ab} + q(1 - \delta_{ab}). \tag{20}$$

This assumption allows us to express $u_a$ given $y$ as

$$u_a = \sigma_y \left(\sqrt{Q - q}\, t_a + \sqrt{q}\, z\right) + y m, \tag{21}$$

where $t_a$ and $z$ are independent standard Gaussian random variables. Hence the integral over $\boldsymbol{x}$ is recast into those over $z, (t_a)_{a=1}^n$. Using the shorthand notation

$$\int_{-\infty}^\infty \frac{dz}{\sqrt{2\pi}}\, e^{-\frac{1}{2}z^2}(\cdots) =: \int Dz\,(\cdots), \tag{22}$$

we have

$$L = \sum_{y=\pm 1} r_y \int Dz \left(\int Dt\, e^{-\beta s_y \ell\left(\sigma_y\left(\sqrt{Q-q}\,t + \sqrt{q}\,z\right) + ym + b,\, y\right)}\right)^n$$

$$= \sum_{y=\pm 1} r_y \int Dz \left(\int Dt\, e^{-\beta s_y \ell(h(t,z,Q,q,m,\sigma_y,yb))}\right)^n, \tag{23}$$

where

$$h(t, z, Q, q, m, \sigma, b) = \sigma\left(\sqrt{Q-q}\,t + \sqrt{q}\,z\right) + m + b, \tag{24}$$

and at the last line of Eq. (23) we used the invariance of the result with respect to (w.r.t.) $z \to -z$, $t \to -t$ and the symmetry of the loss with the single-argument shorthand notation $\ell(yh) = \ell(h, y)$ introduced in sec. 2.2. Equation (23) reveals that $L$ depends on $\{\boldsymbol{w}_a\}_{a=1}^n$ only through $Q = Q(\{\boldsymbol{w}_a\}_{a=1}^n)$, $q = q(\{\boldsymbol{w}_a\}_{a=1}^n)$, and $m = m(\{\boldsymbol{w}_a\}_{a=1}^n)$. Hence

$$[Z^n]_{D^M} = \operatorname{Tr}_{\{\boldsymbol{w}_a\}_{a=1}^n} L^M = \int dQ\, dq\, dm\, V(Q, q, m) L^M(Q, q, m), \tag{25}$$

where we introduced the notation $L(Q, q, m)$ to denote the dependence of $L$ on $Q, q, m$, and where $V(Q, q, m)$ represents the volume of the subshell specified by $Q, q, m$ in the space of $\{\boldsymbol{w}_a\}_{a=1}^n$:

$$V(Q, q, m) := \operatorname{Tr}_{\{\boldsymbol{w}_a\}_{a=1}^n}\left(\prod_{a=1}^n \delta\left(NQ - \|\boldsymbol{w}_a\|^2\right)\delta\left(Nm - \boldsymbol{w}_0^\top \boldsymbol{w}_a\right)\right)\prod_{a<b}\delta\left(Nq - \boldsymbol{w}_a^\top \boldsymbol{w}_b\right). \tag{26}$$

We defer the details of its computation to sec. A and here only show the result:

$$\lim_{n\to 0, N\to\infty} \frac{1}{nN}\log V(1, q, m) = \underset{\{\hat{Q}, \hat{q}, \hat{m}\}}{\operatorname{Extr}}\left\{\frac{1}{2}\hat{Q} + \frac{1}{2}\hat{q}q - \hat{m}m + \frac{1}{2}\log(2\pi) - \frac{1}{2}\log(\hat{Q} + \hat{q}) + \frac{1}{2}\frac{\hat{m}^2 + \hat{q}}{\hat{Q} + \hat{q}}\right\}, \tag{27}$$

where $\text{Extr}_{\{x\}}$ denotes the extremization w.r.t. $x$. This extremization appears as the consequence of the saddle-point/Laplace method which is valid in the limit $N \to \infty$. The volume becomes finite when $Q = 1$ only due to the normalization condition $\|\boldsymbol{w}\|^2 = N$. In the same way we can compute $\lim_{n \to 0, N \to \infty} \frac{1}{nN} \log L$. Combining these terms, the free energy takes the following compact form:

$$-\beta\phi = \lim_{n \to 0, N \to \infty} \frac{1}{nN} \log [Z^n]_{DM}$$

$$= \underset{\{\hat{Q}, \hat{q}, \hat{m}, q, m\}}{\text{Extr}} \left\{ \frac{1}{2}\hat{Q} + \frac{1}{2}\hat{q}q - \hat{m}m + \frac{1}{2}\log(2\pi) - \frac{1}{2}\log(\hat{Q} + \hat{q}) + \frac{1}{2}\frac{\hat{m}^2 + \hat{q}}{\hat{Q} + \hat{q}} \right.$$

$$\left. + \alpha \sum_{y=\pm1} r_y \int Dz \log \left( \int Dt e^{-\beta s_y \ell(h(t,z,1,q,m,\sigma_y,yb))} \right) \right\}, \tag{28}$$

where $\alpha = M/N$. The quantities $\{\hat{Q}, \hat{q}, \hat{m}, q, m\}$ are the order parameters of the present system.

**Zero-temperature limit $\beta \to \infty$** Next we compute the zero-temperature limit $\beta \to \infty$. As discussed in sec. 2.3, the posterior of $\boldsymbol{w}$ concentrates on the set of minimizers of the loss in the limit $\beta \to \infty$. We further expect that the minimizer is unique. In view of this, we adopt the following ansatz for the asymptotic behaviors of the order parameters around the saddle points as $\beta$ becomes large:

$$q = 1 - \chi/\beta, \qquad \chi = O(1), \qquad m = O(1),$$
$$\hat{Q} = -\beta^2\tilde{\chi} + \beta\tilde{Q}, \qquad \hat{q} = \beta^2\tilde{\chi}, \qquad \hat{m} = \beta\tilde{m}. \tag{29}$$

This ansatz will later turn out to be consistent. Inserting these relations into Eq. (28), performing the variable transform $v = t/\sqrt{\beta}$ and taking the limit $\beta \to \infty$ yield

$$u = \lim_{\beta \to \infty} \phi = \underset{\{\tilde{Q}, \tilde{\chi}, \tilde{m}, \chi, m\}}{\text{Extr}} \left\{ -\frac{1}{2}\tilde{Q} + \frac{1}{2}\tilde{\chi}\chi + \tilde{m}m - \frac{1}{2}\frac{\tilde{m}^2 + \tilde{\chi}}{\tilde{Q}} \right.$$

$$\left. - \alpha \sum_{y=\pm1} r_y \int Dz \, G(v_y, h_y, s_y) \right\}. \tag{30}$$

where

$$G(v, h, s) = -\frac{1}{2}v^2 - s\ell(h), \tag{31}$$

$$v_y = \arg\max_v \left\{ -\frac{1}{2}v^2 - s_y\ell(\sigma_y(\sqrt{\chi}v + z) + m + yb) \right\}, \tag{32}$$

$$h_y = \sigma_y(\sqrt{\chi}v_y + z) + m + yb. \tag{33}$$

Here, $v_y$ denotes the maximum point of $G$ w.r.t. the variable $v$ when taking the limit $\beta \to \infty$. Note that the dependence of $v_y$ and $h_y$ on the integration variable $z$ is implicit in the above formulae.

**Equations of state (EOS)** The extremization condition in Eq. (30) yields the following equations:

$$\tilde{Q}^2 = \tilde{m}^2 + \tilde{\chi}, \tag{34a}$$

$$m = \frac{\tilde{m}}{\tilde{Q}}, \tag{34b}$$

$$\chi = \frac{1}{\tilde{Q}}, \tag{34c}$$

$$\tilde{m} = \frac{\alpha}{\sqrt{\chi}} \sum_{y=\pm1} \frac{r_y}{\sigma_y} \int Dz \, v_y, \tag{34d}$$

$$\tilde{\chi} = \alpha \sum_{y=\pm1} r_y \sigma_y^2 \int Dz \, v_y^2. \tag{34e}$$

This set of equations is the EOS for the present problem. An intuitive interpretation of the EOS is given in sec. B. The term $v_y$ appearing in Eqs. (34d) and (34e) is defined via Eq. (32) as a function of $m, \chi, b, y, s_y$. Determining $v_y$ is easy if we are allowed to assume differentiability of the loss $\ell$ on $\mathbb{R}$: introducing the shorthand notation $g = -\frac{d\ell}{dh}$, we find that $v_y$ satisfies

$$v_y = s_y \sigma_y \sqrt{\chi} g\left(h_y\right). \tag{35}$$

It should be noted that $h_y$ appearing on the right-hand side depends on $v_y$ through Eq. (33), so that the above equation is a (non-linear) equation on $v_y$. If there are multiple solutions to Eq. (35), the one yielding the largest value of $G(v_y, h_y, s_y)$ should be selected. Under this differentiability assumption, the extremization condition w.r.t. $m$ can be computed from the partial derivative of $G$ w.r.t. $h$ as $\frac{\partial G(v,h,s)}{\partial h} \frac{\partial h}{\partial m} = sg(h)$: this is because the partial derivative of $G$ w.r.t. $v$ vanishes since $v$ is fixed at the extremum value of $G$ given $h$ and $s$ as shown in Eq. (32). The term $sg(h)$ can be rewritten using Eq. (35), yielding the right-hand side of Eq. (34d). A similar rewriting using Eq. (35) is done for the extremization condition w.r.t. $\chi$, yielding the right-hand side of Eq. (34e). Although the differentiability assumption does not hold for some losses having singularities such as the zero-one loss, those losses are usually representable as a limit of a certain differentiable function: for example, the zero-one loss can be expressed as

$$\ell_{01}(h) := \left\{ \begin{array}{ll} 1 & (h < 0) \\ 0 & (h \geq 0) \end{array} \right\} = \lim_{\gamma \to \infty} \frac{1}{2}\left(1 - \tanh(\gamma h)\right) =: \lim_{\gamma \to \infty} \ell_{01,\gamma}(h). \tag{36}$$

Hence, for such losses with singularities, the above discussion should be interpreted as that for such smoothed versions of the losses, and the smoothness-controlling parameter ($\gamma$ in Eq. (36)) is sent to an appropriate limit after the computation. The resultant formula (34) still holds after the limit even when the derivative $g$ itself is not meaningful in the limit.

The EOS constitutes the basis of the following study. A special focus is on the overlap $m = \left\langle \boldsymbol{w}^\top \boldsymbol{w}_0 \right\rangle / N$, because it characterizes the performance of the feature learning as discussed in sec. 2.2.

### 3.3 Behaviors of quantities of interest

By specifying the functional form of the loss $\ell$, one can obtain the values of the order parameters and the energy, via computing $v_\pm$ with the loss and numerically solving the EOS. We consider two choices for the loss $\ell$ as representative examples: the zero-one loss with the perceptron $\ell_{01\text{pe}}(h, y) = (1 - \text{sgn}(yh))/2$ and the CE loss with the logit function $\ell_{\text{CElo}}(h, y) = -hy + \log(2\cosh(h))$. Hereafter these two losses are referred to as 01pe and CElo, respectively. The aim here is to investigate how the order parameters depend on the parameters in order to assess the feature learning performance.

### 3.3.1 Equivariance case $(\sigma_+^2 = \sigma_-^2)$

We start from a desirable situation where the two class variances are equal: $\sigma_+^2 = \sigma_-^2 =: \sigma^2$. As examples, we compare the balanced case $r_+ = 0.5$ and an imbalanced case $r_+ = 0.2$, with $\sigma = 0.6$. For illustration of these cases, the probability density functions (PDF) of $\boldsymbol{x}$ projected onto $\boldsymbol{w}_0/\sqrt{N}$ are plotted in Fig. 1. Besides, we fix $\alpha = 20$ in the following plots unless otherwise stated; the results for other values of $\alpha$, if not too small, were qualitatively similar as far as we have checked.

Firstly, we show $m$ and $u$ plotted against $b$ for $s_+ = 0.1, 0.5, 0.9$: the results for $r_+ = 0.5$ and $0.2$ are shown in Fig. 2 and Fig. 3, respectively. It should be noted that the class centers are located at $\pm \boldsymbol{w}_0/\sqrt{N}$ in our problem setting, so that a reasonable choice of the parameter $b$ in view of the task of classification would intuitively be $b \in [-1, 1]$: indeed, if otherwise, the two class centers $\pm \boldsymbol{w}_0/\sqrt{N}$ are located on the same side of the decision plane $\boldsymbol{w}^\top \boldsymbol{x}/\sqrt{N} + b = 0$. In the following, we nevertheless investigate the behaviors of the models over wider ranges of $b$. Fig. 2 (a) and Fig. 3 (a) are the results with 01pe, whereas Fig. 2 (b) and Fig. 3 (b) are those with CElo. As observed in Fig. 2, the $b$-dependence of $m$ in the balanced case is very different between 01pe and CElo: with 01pe, it is almost symmetric w.r.t. $b$ even with the strong reweighting factors $s_+ = 0.1$ and $0.9$, while it exhibits clear asymmetry with CElo. The values of $b$ at which $m$ achieves its maximum are also different: $m$ achieves its maximum around $b \approx \pm 1$ with 01pe, while with CElo the

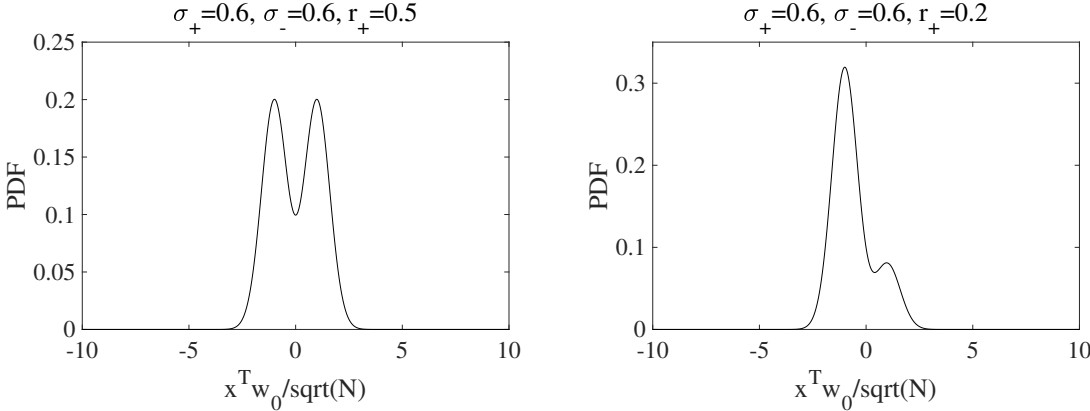

Figure 1: PDFs of $\boldsymbol{x}$ projected onto $\boldsymbol{w}_0$ with $\sigma = 0.6$ for $r_+ = 0.5$ (left) and 0.2 (right).

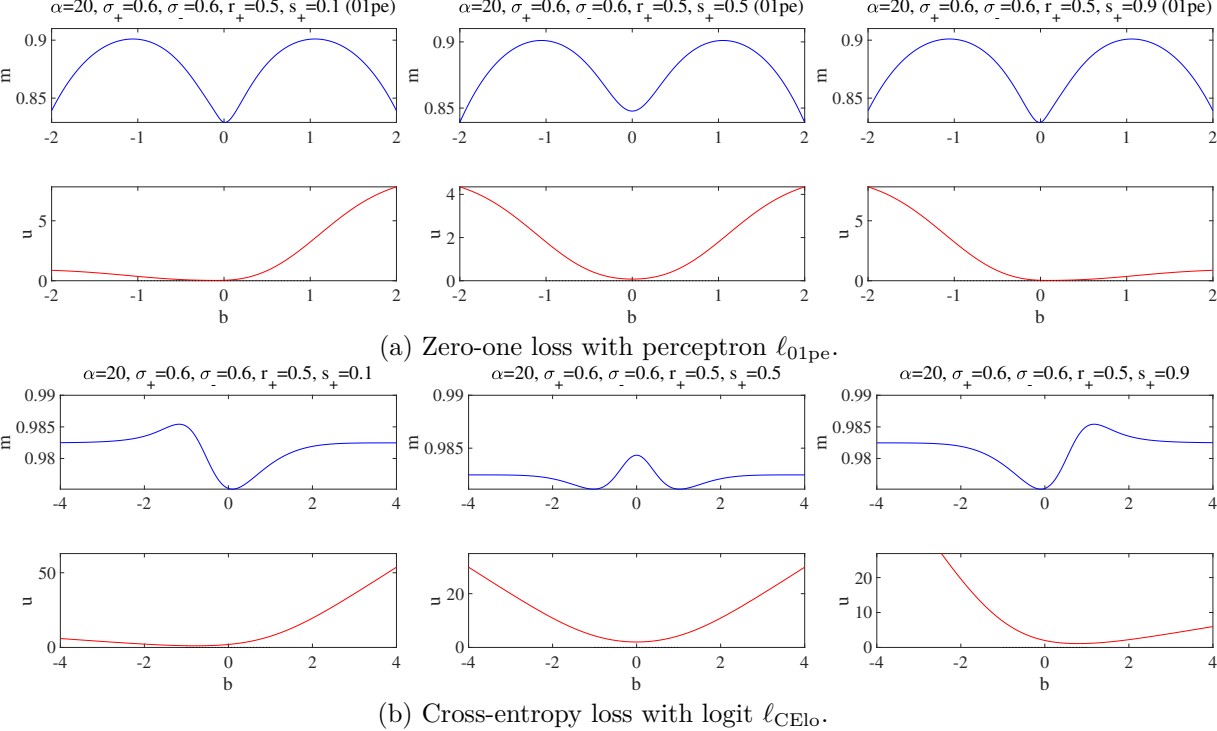

(a) Zero-one loss with perceptron $\ell_{01\text{pe}}$.

(b) Cross-entropy loss with logit $\ell_{\text{CElo}}$.

Figure 2: Plots of $m$ and $u$ against $b$ in the balanced case $r_+ = 0.5$ for $s_+ = 0.1$ (left), 0.5 (middle), and 0.9 (right). (a) Zero-one loss with perceptron $\ell_{01\text{pe}}$. (b) Cross-entropy loss with logit $\ell_{\text{CElo}}$.

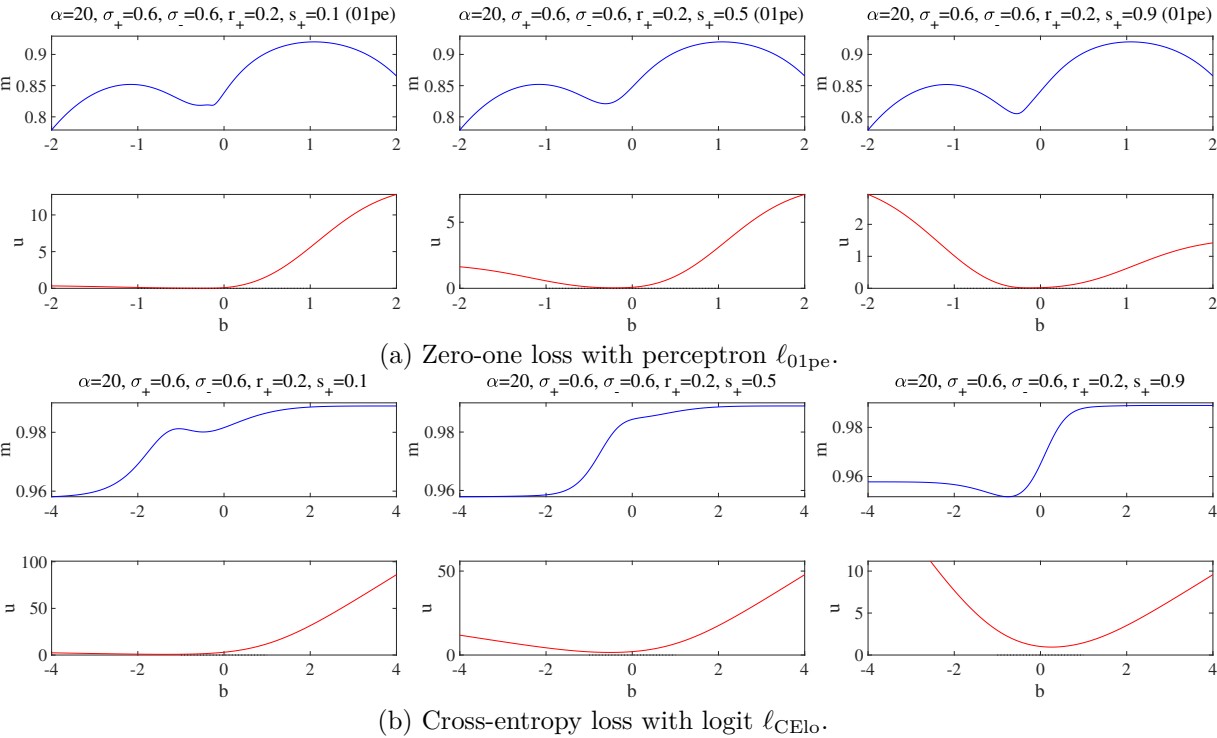

(a) Zero-one loss with perceptron $\ell_{01\mathrm{pe}}$.

(b) Cross-entropy loss with logit $\ell_{\mathrm{CElo}}$.

Figure 3: Plots of $m$ and $u$ against $b$ in the imbalanced case $r_+ = 0.2$ for $s_+ = 0.1$ (left), 0.5 (middle), and 0.9 (right). (a) Zero-one loss with perceptron $\ell_{01\mathrm{pe}}$. (b) Cross-entropy loss with logit $\ell_{\mathrm{CElo}}$.

maximum is achieved at $b$ less than $-1$ for $s_+ = 0.1$, at $b = 0$ for $s_+ = 0.5$, and at $b$ larger than 1 for $s_+ = 0.9$. Meanwhile, as observed in Fig. 3 for the imbalanced case $r_+ = 0.2$, the maximum of $m$ seems to be obtained at a large positive $b$ in all the cases: this is considered to be natural since the positive bias enhances the probability of the minority class at $r_+ = 0.2$. As an overall tendency, the value of $m$ tends to be larger with CElo than that with 01pe, suggesting the superiority of the CE loss in the feature learning.

Secondly, to examine the maximum performance of feature learning, we compute the maximum of $m$ against $b$, $m_{\max} = \max_b m(b)$, and plot it against $s_+$ in Fig. 4. Fig. 4 (a) and (b) are with 01pe and CElo, respectively; the left and right columns are the plots with $r_+ = 0.5$ and 0.2, respectively. The maximum location $b(m_{\max}) = \arg\max_b m(b)$ and the corresponding energy value $u(m_{\max}) = u(b(m_{\max}))$ are also plotted. An interesting observation is that with 01pe the maximum of $m_{\max}$ is obtained at the no-reweighting situation $s_+ = 1/2$ even in the imbalanced case (Fig. 4 (a), right), while with CElo the $m$'s maximum is located at some value different from $s_+ = 1/2$ even in the balanced case $r_+ = 0.5$ (Fig. 4 (b), left). This property of 01pe persisted as far as we have numerically investigated. This may be related to Kang et al.'s observation, although 01pe is not typically used in practical situations.

In practical situations one cannot directly maximize the overlap $m$ since one does not know $\boldsymbol{w}_0$. One will instead minimize the loss to obtain a reasonable estimator. From this viewpoint, thirdly, the minimum of $u$ w.r.t. $b$, $u_{\min} = \min_b u(b)$, is plotted against $s_+$ in Fig. 5, with the minimum location $b(u_{\min}) = \arg\min_b u(b)$ and the corresponding overlap value $m(u_{\min}) = m(b(u_{\min}))$. This figure shows the intriguing behavior of $m(u_{\min})$: in the balanced case $r_+ = 0.5$, its curve is symmetric around $s_+ = 0.5$ and the maximum is obtained at $s_+ = 0.5$ as expected, with both 01pe and CElo. In the imbalanced case $r_+ = 0.2$, however, the tendency is different between 01pe and CElo: with 01pe, there is a maximum at a certain value of $s_+$ greater than 0.5, which is natural because the values of $s_+$ greater than 0.5 enhance the probability of the minority (positive here) class. With CElo, on the other hand, the opposite occurs and the overlap maximum is obtained at $s_+ = 0$, meaning the complete disregard of the minority class, and we numerically confirmed

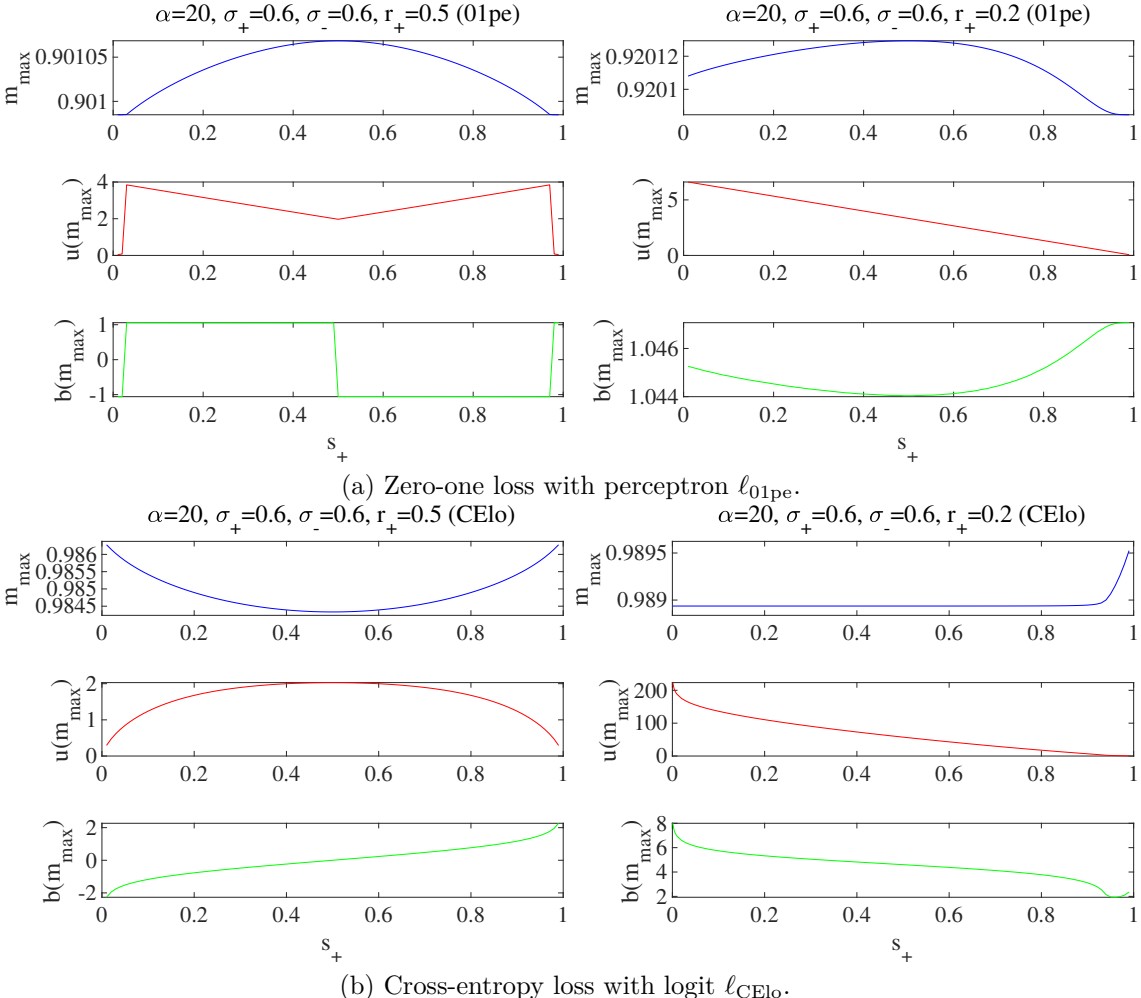

(a) Zero-one loss with perceptron $\ell_{01\text{pe}}$.

(b) Cross-entropy loss with logit $\ell_{\text{CElo}}$.

Figure 4: Plots of $m_{\max}, u(m_{\max}), b(m_{\max})$ against $s_+$ for $r_+ = 0.5$ (left) and $r_+ = 0.2$ (right). (a) Zero-one loss with perceptron $\ell_{01\text{pe}}$. (b) Cross-entropy loss with logit $\ell_{\text{CElo}}$.

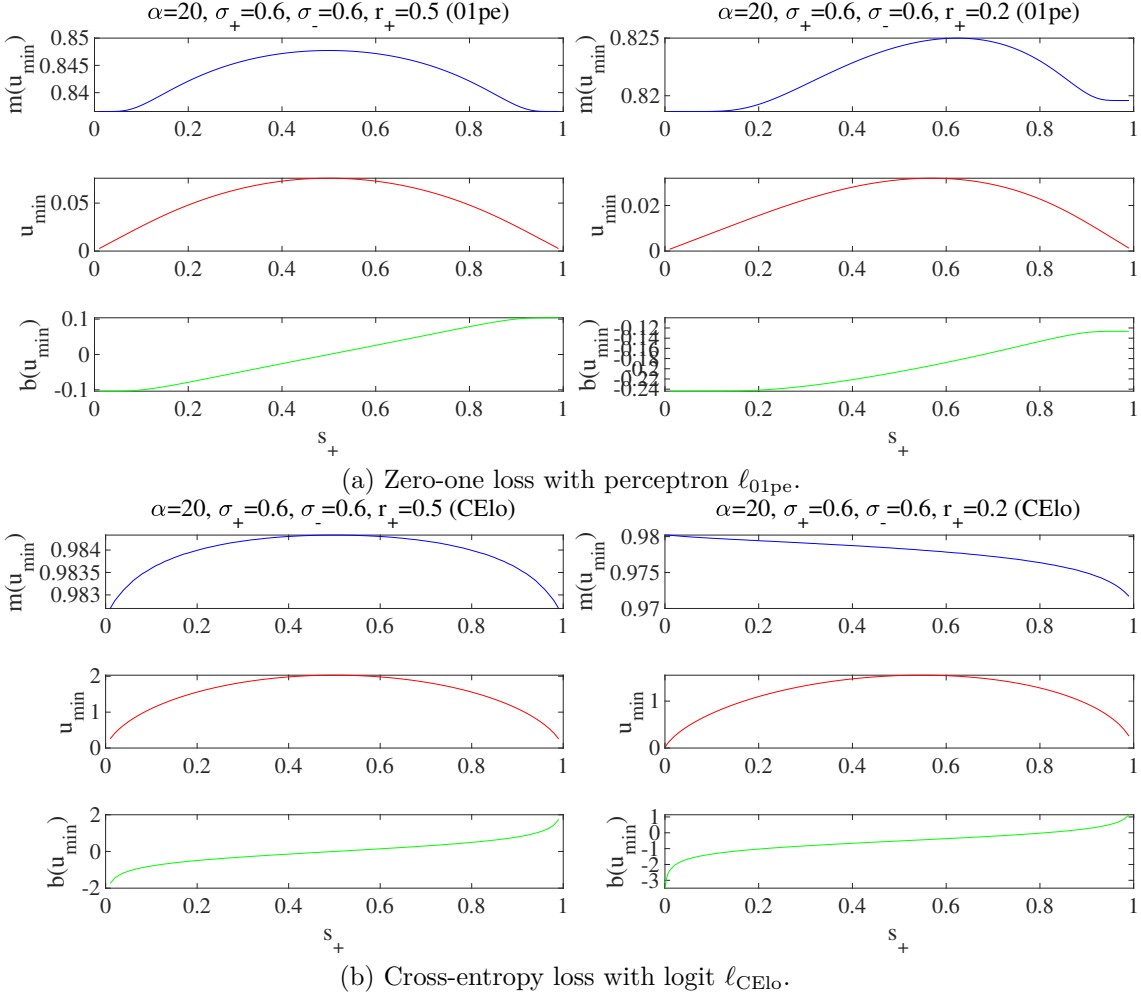

(a) Zero-one loss with perceptron $\ell_{01\text{pe}}$.

(b) Cross-entropy loss with logit $\ell_{\text{CElo}}$.

Figure 5: Plots of $m(u_{\min}), u_{\min}, b(u_{\min})$ against $s_+$ for $r_+ = 0.5$ (left) and $r_+ = 0.2$ (right). (a) Zero-one loss with perceptron $\ell_{01\text{pe}}$. (b) Cross-entropy loss with logit $\ell_{\text{CElo}}$.

that the corresponding minimum location $b(u_{\min})$ goes to $-\infty$. This is rather counterintuitive, and provides a lesson that it is not straightforward to predict how the interplay among the loss, classifier, and reweighting factor would influence the feature learning performance.

In recent practices, the bias $b$ is occasionally neglected (Cao et al., 2019; Menon et al., 2020; Kini et al., 2021). This is because DNNs learn feature vectors also from the data, and hence the adjustment of the feature space origin, that is the effect of the bias, can be incorporated by the learning even without the bias. Accordingly, we lastly examine the case $b = 0$, which is actually a natural choice since the origin is located at the middle point of the two class centers in our setting. The overlap and energy in this case are plotted against $s_+$ in Fig. 6. A very interesting observation is that the maximum of $m$ is achieved at the

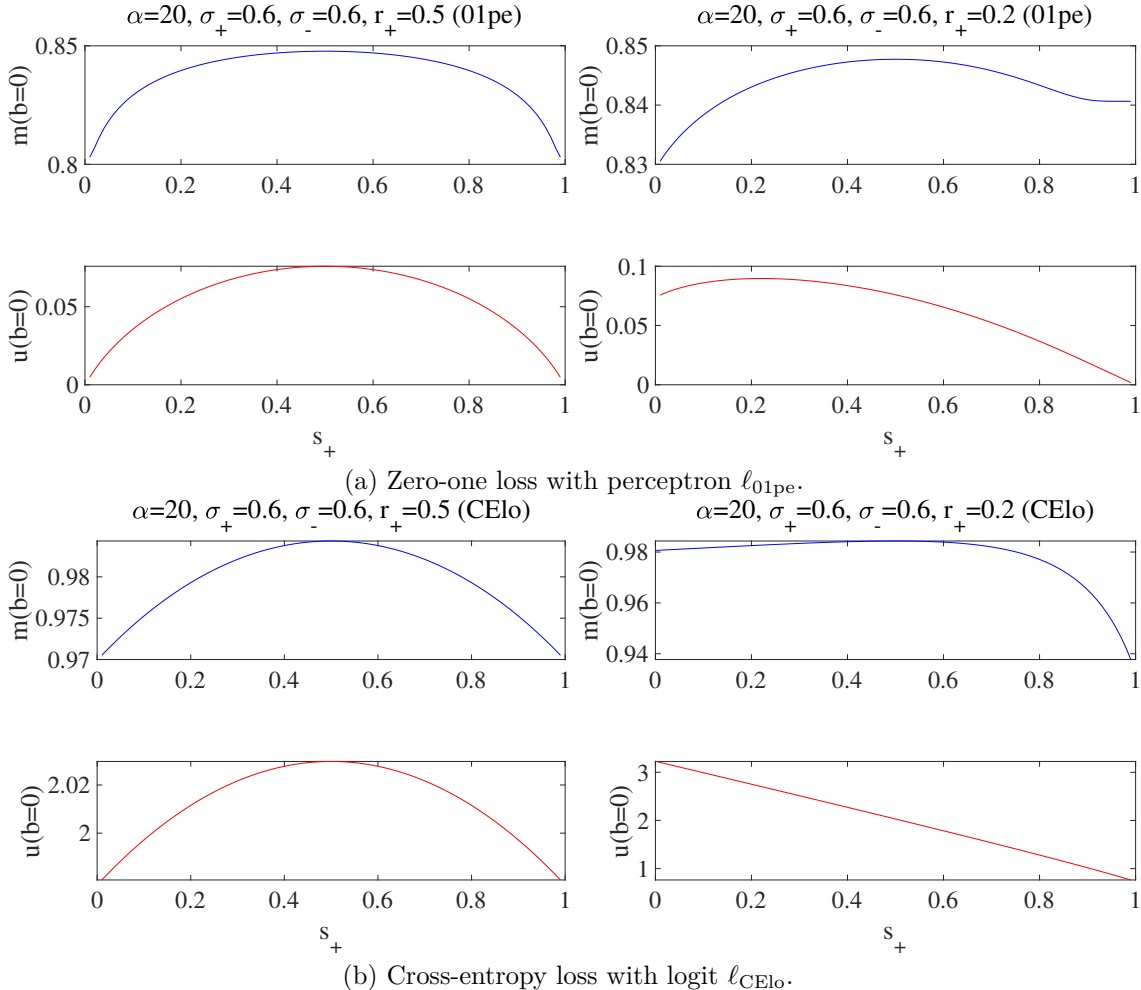

(a) Zero-one loss with perceptron $\ell_{01\mathrm{pe}}$.

(b) Cross-entropy loss with logit $\ell_{\mathrm{CElo}}$.

Figure 6: Plots of $m$ and $u$ at $b = 0$ against $s_+$ for $r_+ = 0.5$ (left) and $r_+ = 0.2$ (right). (a) Zero-one loss with perceptron $\ell_{01\mathrm{pe}}$ and (b) Cross-entropy loss with logit $\ell_{\mathrm{CElo}}$. A crucial observation is that the maximum of $m$ is always obtained at the no reweighting situation $s_+ = 1/2$, irrespectively of the models, losses, and the degree of class imbalance.

no-reweighting situation $s_+ = 0.5$ in all the cases. Actually, this property holds irrespectively of the loss, model, or the degree of class imbalance: we will provide a strong analytical evidence of this fact later in sec. 3.4. This property means that the feature learning has its best performance when no resampling/reweighting is applied if the bias is appropriately chosen to make the decision boundary to be located equidistantly from the two class centers. This provides an analytical support for Kang et al.'s observation and constitutes one of our main results in this paper.

To summarize, we enumerate our findings in sec. 3.3.1:

1. The overlap value $m$ tends to be larger with CElo than with 01pe.

2. The values of $b$ with which maximum values of $m$ are achieved tend to be strongly dependent on the choice of the model and the loss.

3. The maximum overlap $m_{\max} = \max_b m(b)$ also shows a strong dependence on the choice of the model and the loss. With 01pe, $m_{\max}$ takes its largest value at the no-reweighting situation $s_+ = 0.5$, whereas with CElo it is obtained at some extreme values of $s_+$.

4. The overlap $m(u_{\min})$ at the point of minimizing the loss $u$ shows a moderate dependence on $s_+$. The dependence on $s_+$ is natural with 01pe but is counterintuitive with CElo. With CElo, the maximum is at $s_+ = 0.5$ in the balanced case but is at the extreme value of $s_+$ in the imbalanced case.

5. The overlap $m$ takes its maximum at the no-reweighting situation $s_+ = 0.5$ irrespectively of the loss, model, or degree of class imbalance when the bias is appropriately chosen to make the decision boundary to be located equidistantly from the two class centers.

The assumption on the bias in item 5 of the above list is what is considered desirable in the standard view of classification. Hence, Kang et al.'s observation is a property that holds widely when features and bias are set to be in such a desirable situation, which conversely implies that their learning works well. Meanwhile, our other findings such as item 4 in the above suggest some other bias values different from the desirable one; the resultantly selected value tends to take an extraordinary value outside the reasonable range $[-1, 1]$ of the bias $b$ in the present setting. Presumably, this has prevented researchers from examining such extreme bias values in practical situations, and it may be an interesting future work to study such extreme biases in real-world datasets.

### 3.3.2 Non-equivariance case

We turn to the non-equivariance case. As an example, we examine $\sigma_+ = 1, \sigma_- = 0.5$ with $r_+ = 0.5$ and $0.2$ as depicted in Fig. 7. We compare the result for this case with the one in the equivariance case, especially

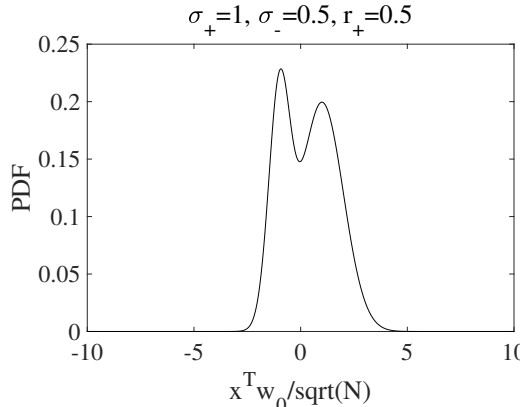 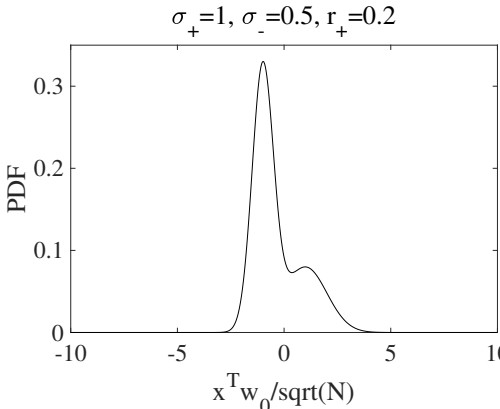

Figure 7: PDFs for $\sigma_+ = 1, \sigma_- = 0.5$ at $r_+ = 0.5$ (left) and $r_+ = 0.2$ (right).

focusing on the reweighting factor dependence of $m$ after erasing the bias dependence as in Figs. 4–6.

Figure 8 is the counterpart of Fig. 4 in which $m_{\max}$ and the related quantities are plotted against $s_+$. As expected from the asymmetry between the classes, the maximum location of $m_{\max}$ is not $s_+ = 1/2$ anymore with either 01pe or CElo. Another interesting observation is that the maximum location is $s_+ < 1/2$ with 01pe and $s_+ > 1/2$ with CElo, showing that the effect of reweighting on the feature learning is not simple.

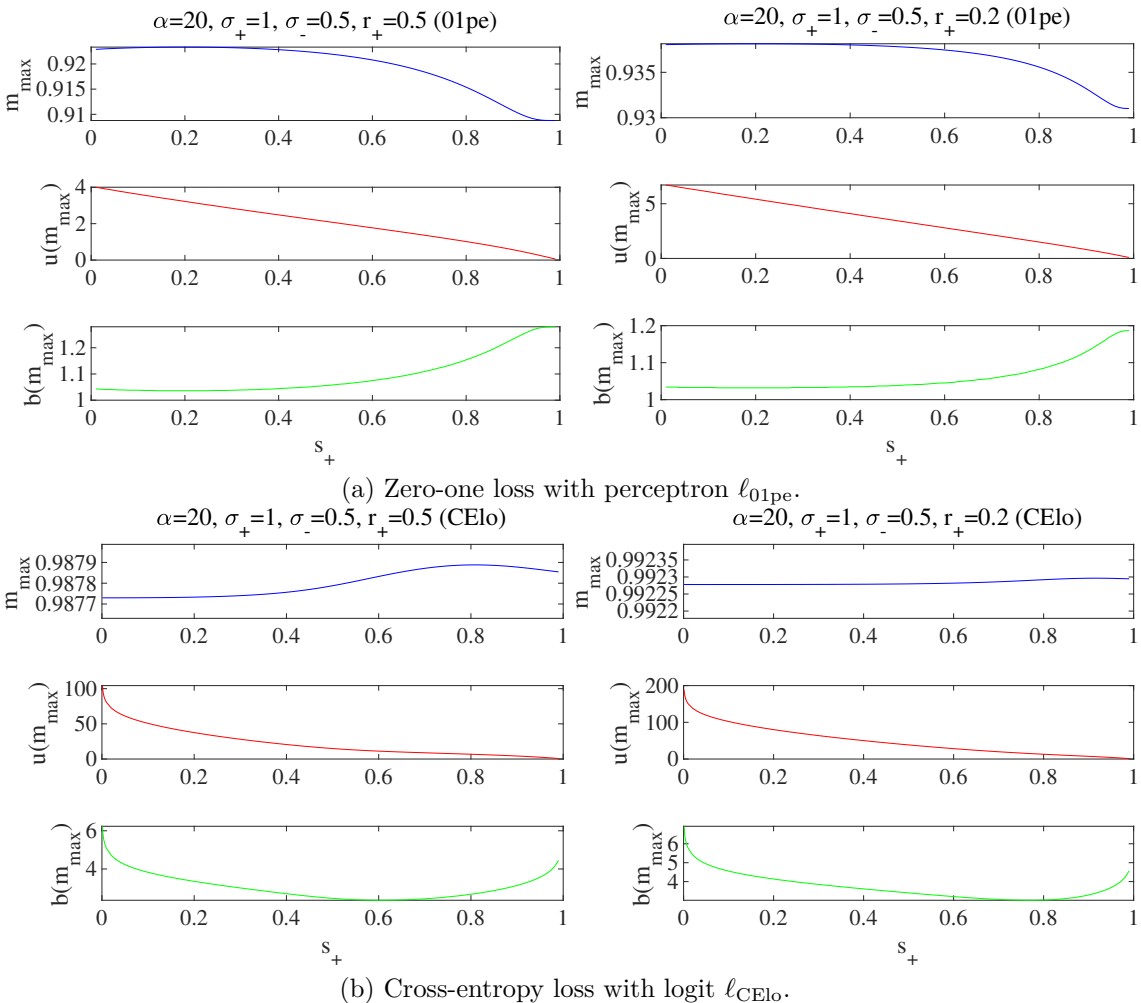

(a) Zero-one loss with perceptron $\ell_{01\text{pe}}$.

(b) Cross-entropy loss with logit $\ell_{\text{CElo}}$.

Figure 8: Plots of $m_{\max}, u(m_{\max}), b(m_{\max})$ against $s_+$ for $r_+ = 0.5$ (left) and $r_+ = 0.2$ (right) in the non-equivariance case. (a) Zero-one loss with perceptron $\ell_{01\text{pe}}$. (b) Cross-entropy loss with logit $\ell_{\text{CElo}}$.

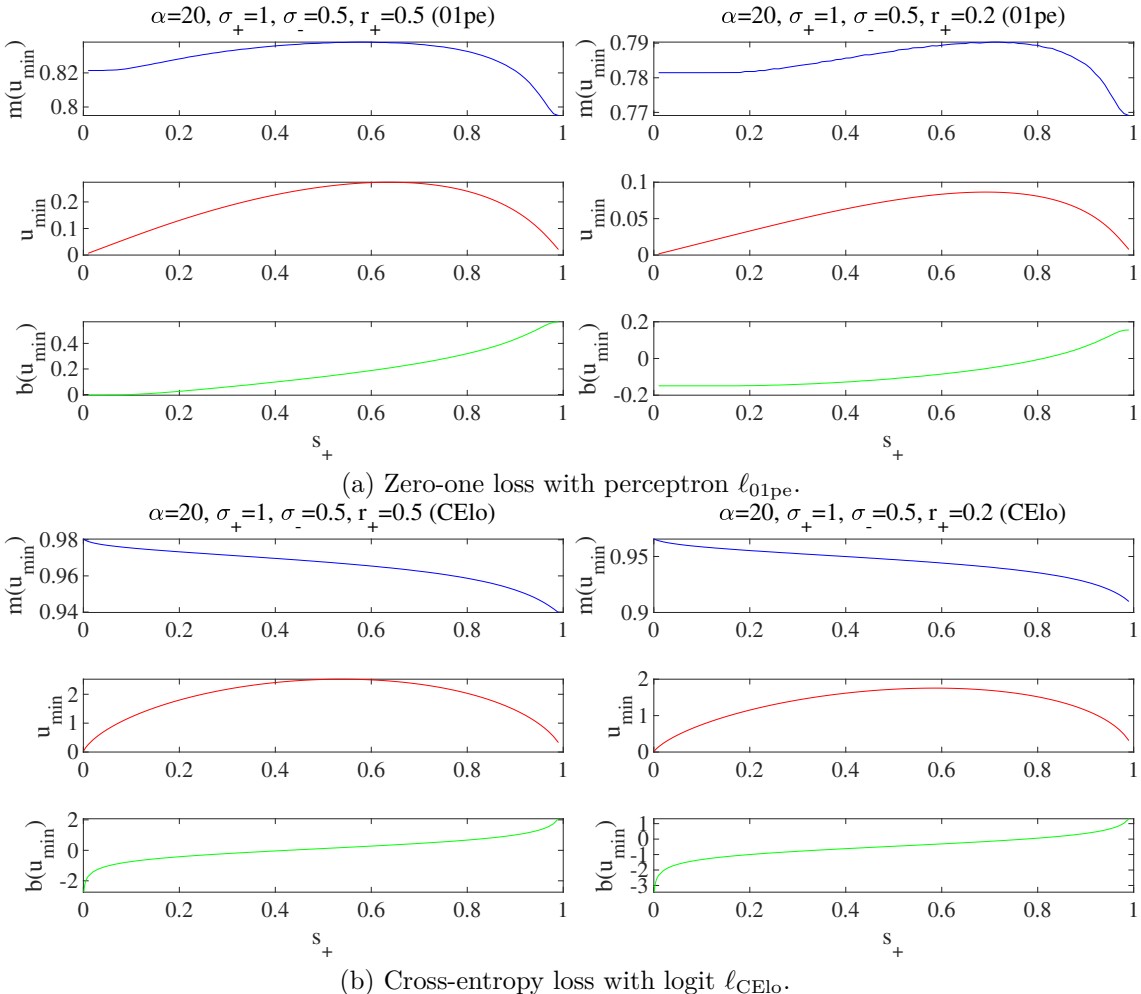

(a) Zero-one loss with perceptron $\ell_{01\mathrm{pe}}$.

(b) Cross-entropy loss with logit $\ell_{\mathrm{CElo}}$.

Figure 9: Plots of $m(u_{\min}), u_{\min}, b(u_{\min})$ against $s_+$ for $r_+ = 0.5$ (left) and $r_+ = 0.2$ (right) in the non-equivariance case. (a) Zero-one loss with perceptron $\ell_{01\mathrm{pe}}$. (b) Cross-entropy loss with logit $\ell_{\mathrm{CElo}}$.

Next, we study the loss-minimizing result in Fig. 9, which is the counterpart of Fig. 5. This time, again due to the asymmetry, the maximum location of $m$ is not at $s_+ = 1/2$ in all the cases. It is at $s_+ > 1/2$ with 01pe and at $s_+ = 0$ with CElo, which is in contrast to the overlap-maximizing result in Fig. 8. Looking at both of the equivariance and non-equivariance results with CElo shown in Figs. 5 and 9, respectively, we see that the asymmetry due to the imbalance both in the number of examples and the variance magnitude commonly leads to extreme values of $s_+$ for the best feature learning.

Finally, we examine the no-bias case $b = 0$ in Fig. 10. Similarly to the above two cases, the maximum of $m$

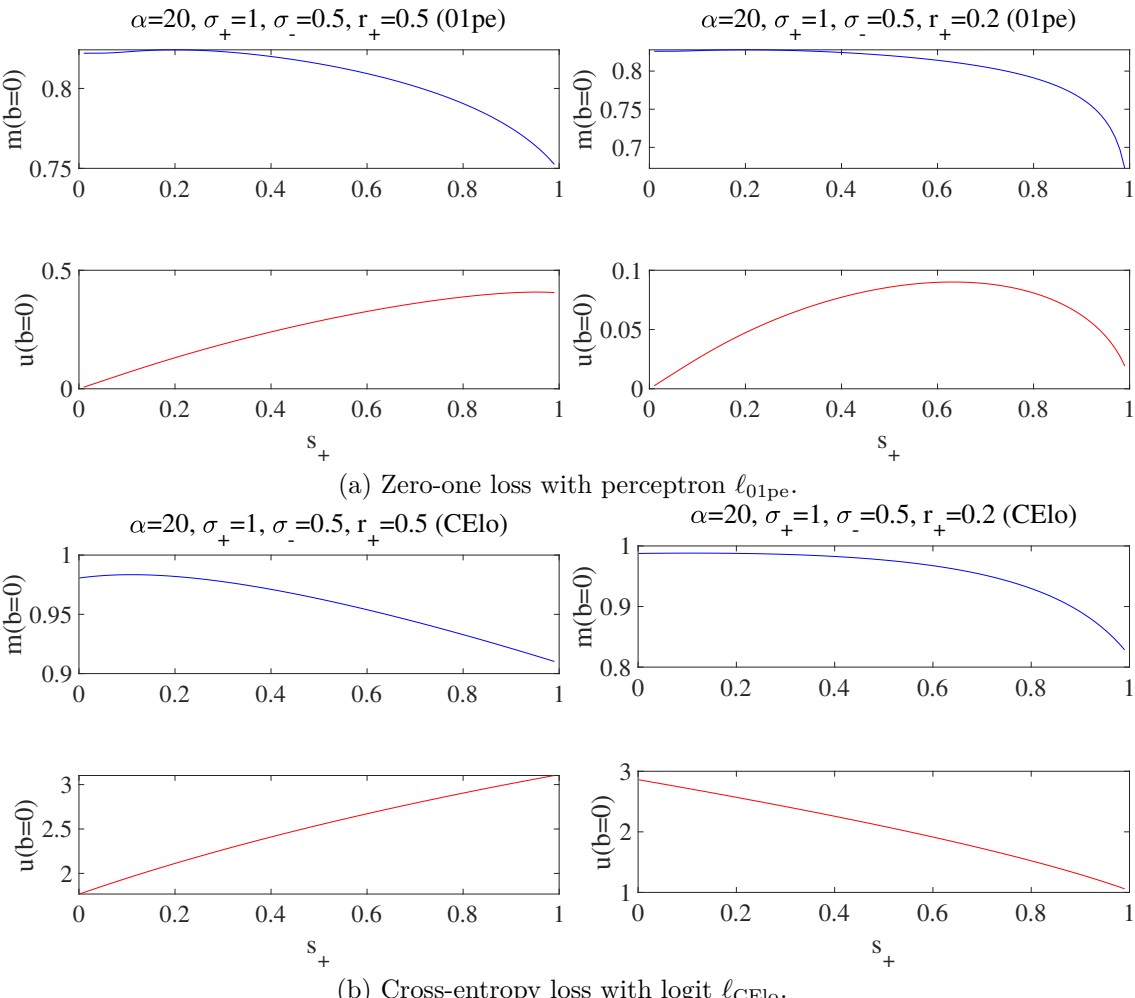

(a) Zero-one loss with perceptron $\ell_{01\text{pe}}$.

(b) Cross-entropy loss with logit $\ell_{\text{CElo}}$.

Figure 10: Plots of $m(b = 0), u(b = 0)$ against $s_+$ for $r_+ = 0.5$ (left) and $r_+ = 0.2$ (right) in the non-equivariance case. (a) Zero-one loss with perceptron $\ell_{01\text{pe}}$. (b) Cross-entropy loss with logit $\ell_{\text{CElo}}$.

is not at $s_+ = 1/2$. An interesting observation is that the maximum is located commonly in $s_+ < 1/2$. This may not be surprising since the class with $y = -1$ has a smaller variance and thus is considered to carry more information about the feature direction.

Overall, in the non-equivariance case, we found no empirical evidence supporting Kang et al.'s observation. This implies that the equivariance condition of the feature across the two classes may be an important ingredient for their observation. This condition may be achieved as a result of feature learning in DNNs. Actually, a recent paper reported that an interesting phenomenon widely occurs in successful DNNs for classification (Papyan et al., 2020). This phenomenon is called neural collapse (NC) and its important ingredient is *within-class variation collapse*, meaning that the feature vectors of the same class converge to an identical vector as a result of learning. This is an example of the equivariance condition with $\sigma_+, \sigma_- \to 0$.

The mechanism why NC occurs is also understood from a simple model analysis (Fang et al., 2021). These partially support the practical relevance of our present results.

### 3.4 Analytical evidence that the maximum value of $m$ is achieved at $s_+ = 1/2$

In this subsection, we provide an analytical evidence for the observation that $m$ takes its maximum at the no-reweighting situation $s_+ = 1/2$ irrespectively of the loss, model, or degree of class imbalance when the feature vector distributions of the two classes are equivariance and the bias is appropriately chosen to make the decision boundary to be located equidistantly from the two class centers. The last two conditions mean $\sigma_+ = \sigma_- (=: \sigma)$ and $b = 0$ in the present setting, and we assume them throughout this subsection.

What we will show below is that $s_+ = 1/2$ yields an extremum of $m$ under the above assumptions. To show this, we compute the derivatives of the order parameters w.r.t. $s_+$. For notational convenience, we write the derivative of a quantity $A$ w.r.t. $s_+$ as $\dot{A} := \frac{dA}{ds_+}$, and the order parameter vectors as $\tilde{\boldsymbol{\Omega}} = (\tilde{m}, \tilde{\chi})^\top$ and $\boldsymbol{\Omega} = (m, \chi)^\top$.

Taking the derivatives w.r.t. $s_+$ of the EOS (34), we can derive a set of linear equations of $\dot{\tilde{\boldsymbol{\Omega}}}, \dot{\boldsymbol{\Omega}}$. For example, the derivative of Eq. (34a) yields

$$\dot{\tilde{Q}} = \frac{\tilde{m}}{\tilde{Q}}\dot{\tilde{m}} + \frac{1}{2\tilde{Q}}\dot{\tilde{\chi}}. \tag{37}$$

Similarly, taking the derivative of eqs. (34b) and (34c) and rewriting $\dot{\tilde{Q}}$ by using Eq. (37), we have a relation transforming $\dot{\tilde{\boldsymbol{\Omega}}}$ to $\dot{\boldsymbol{\Omega}}$ as

$$\dot{\boldsymbol{\Omega}} = \tilde{T}\dot{\tilde{\boldsymbol{\Omega}}}, \tag{38a}$$

$$\tilde{T} = \frac{1}{\tilde{Q}^3}\begin{pmatrix} \tilde{\chi} & -\frac{1}{2}\tilde{m} \\ -\tilde{m} & -\frac{1}{2} \end{pmatrix}. \tag{38b}$$

In the same way, after lengthy algebras, the derivatives of eqs. (34d) and (34e) lead to another relation transforming $\dot{\boldsymbol{\Omega}}$ to $\dot{\tilde{\boldsymbol{\Omega}}}$:

$$\dot{\tilde{\boldsymbol{\Omega}}} = \boldsymbol{c} + T\dot{\boldsymbol{\Omega}}, \tag{39}$$

where

$$g_y := g(h_y), \tag{40a}$$

$$g_y' := \frac{dg(h)}{dh}\Big|_{h=h_y}, \tag{40b}$$

$$D_y := 1 - \sigma\chi s_y g_y', \tag{40c}$$

$$\boldsymbol{c} = \alpha\begin{pmatrix} \sum_{y=\pm 1} y r_y \int Dz\, \frac{g_y}{D_y} \\ \sum_{y=\pm 1} y r_y s_y \int Dz\, \frac{g_y^2}{D_y} \end{pmatrix}, \tag{40d}$$

$$T = \begin{pmatrix} T_{11} & T_{12} \\ T_{21} & T_{22} \end{pmatrix}, \tag{40e}$$

$$T_{11} = \alpha\left(\sum_{y=\pm 1} r_y s_y \int Dz\, \frac{g_y'}{D_y}\right), \tag{40f}$$

$$T_{12} = \sigma^2\alpha\left(\sum_{y=\pm 1} r_y s_y^2 \int Dz\, \frac{g_y g_y'}{D_y}\right), \tag{40g}$$

$$T_{21} = 2\sigma^2\alpha\left(\sum_{y=\pm 1} r_y s_y^2 \int Dz\, \frac{g_y g_y'}{D_y}\right) (= 2T_{12}), \tag{40h}$$

$$T_{22} = 2\sigma^2\alpha\left(\sum_{y=\pm 1} r_y \sigma^2 s_y^3 \int Dz\, \frac{g_y^2 g_y'}{D_y}\right). \tag{40i}$$

Note that the expressions in Eq. (40) assume that the loss $\ell$ is twice differentiable on $\mathbb{R}$. For losses having singularities such as the zero-one loss, those expressions should be interpreted as appropriate limits of those for their smoothed versions as discussed in sec. 3.2. The smoothness-controlling parameter ($\gamma$ in Eq. (36)) can be arbitrary because the following discussion holds irrespectively of its value.

Using the identity $g'_y/D_y = \left(-1 + \frac{1}{D_y}\right)/(\sigma^2 \chi s_y)$, we rewrite Eq. (40) as

$$T_{11} = \frac{\alpha}{\sigma^2 \chi}\left(-1 + \sum_{y=\pm 1} r_y \int Dz\, \frac{1}{D_y}\right), \tag{41}$$

$$T_{12} = \frac{\alpha}{\chi}\left(-\sum_{y=\pm 1} r_y s_y \int Dz\, g_y + \sum_{y=\pm 1} r_y s_y \int Dz\, \frac{g_y}{D_y}\right), \tag{42}$$

$$T_{21} = 2T_{12}, \tag{43}$$

$$T_{22} = \frac{2\sigma^2 \alpha}{\chi}\left(-\sum_{y=\pm 1} r_y s_y^2 \int Dz\, g_y^2 + \sum_{y=\pm 1} r_y s_y^2 \int Dz\, \frac{g_y^2}{D_y}\right). \tag{44}$$

Putting eqs. (38a) and (39) together, we obtain the equation to $\dot{\boldsymbol{\Omega}}$:

$$\dot{\boldsymbol{\Omega}} = \tilde{T}T\boldsymbol{c} + \tilde{T}T\dot{\boldsymbol{\Omega}} =: \boldsymbol{b} + A\dot{\boldsymbol{\Omega}}. \tag{45}$$

If we assume the extremization condition $\dot{m} = \dot{\Omega}_1 = 0$, the following relation can be derived:

$$-\frac{b_1}{A_{12}} = \frac{b_2}{1 - A_{22}}(= \dot{\chi}). \tag{46}$$

Our discussion completes by showing that this condition is satisfied if $s_+ = s_- = 1/2 =: s$. In the situation $s_+ = s_- = s$ with $b = 0$, the symmetry $g_+ = g_- =: g$, $D_+ = D_- =: D$ holds. This simplifies many terms:

$$\tilde{m} \to \alpha s \int Dz\, g, \tag{47a}$$

$$\tilde{\chi} \to \sigma^2 \alpha s^2 \int Dz\, g^2, \tag{47b}$$

$$c_1 \to \alpha(2r_+ - 1) \int Dz\, \frac{g}{D}, \tag{47c}$$

$$c_2 \to 2\sigma^2 \alpha s(2r_+ - 1) \int Dz\, \frac{g^2}{D}, \tag{47d}$$

$$T_{11} \to \frac{\alpha}{\sigma^2 \chi}\left(-1 + \int Dz\, \frac{1}{D}\right), \tag{47e}$$

$$T_{12} \to \frac{\alpha}{\chi}\left(-\int Dz\, g' + \int Dz\, \frac{g'}{D}\right), \tag{47f}$$

$$T_{21} = 2T_{12}, \tag{47g}$$

$$T_{22} \to \frac{2\sigma^2 \alpha}{\chi}\left(-\int Dz\,(g')^2 + \int Dz\, \frac{(g')^2}{D}\right). \tag{47h}$$

Using these, we obtain

$$b_1 = \tilde{T}_{11}c_1 + \tilde{T}_{12}c_2 \to \frac{\sigma^2 \alpha^2 s^2 (2r_+ - 1)}{\tilde{Q}^3}\left\{\int Dz\, g^2 \int Dz\, \frac{g}{D} - \int Dz\, g \int Dz\, \frac{g^2}{D}\right\}, \tag{48}$$

$$b_2 = \tilde{T}_{21}c_1 + \tilde{T}_{22}c_2 \to -\frac{\alpha s(2r_+ - 1)}{\tilde{Q}^3}\left\{\alpha \int Dz\, g \int Dz\, \frac{g}{D} + \sigma^2 \int Dz\, \frac{g^2}{D}\right\}, \tag{49}$$

$$A_{12} = \tilde{T}_{11}T_{12} + \tilde{T}_{12}T_{22} \to \frac{\sigma^2 \alpha^2 s^3}{\tilde{Q}^2}\left\{\int Dz\, g^2 \int Dz\, \frac{g}{D} - \int Dz\, g \int Dz\, \frac{g^2}{D}\right\}, \tag{50}$$

$$A_{22} = \tilde{T}_{21}T_{12} + \tilde{T}_{22}T_{22} \to 1 - \frac{\alpha s^2}{\tilde{Q}^2}\left\{\left(\alpha \int Dz\, g \int Dz\, \frac{g}{D} + \sigma^2 \int Dz\, \frac{g^2}{D}\right)\right\}. \tag{51}$$

In the transformations above, we have used the EOS relation $\chi = \tilde{Q}^{-1}$ and

$$\tilde{Q}^2 = \tilde{m}^2 + \tilde{\chi} \to \alpha^2 s^2 \int Dz \, g^2 + \sigma^2 \alpha s^2 \int Dz \, g^2 \Rightarrow \frac{\tilde{Q}^2}{\alpha s^2} \to \alpha \int Dz \, g^2 + \sigma^2 \int Dz \, g^2. \tag{52}$$

Then, we finally arrive at

$$\frac{b_1}{A_{12}} = \frac{2r_+ - 1}{\tilde{Q}s} = -\frac{b_2}{1 - A_{22}}, \tag{53}$$

showing that Eq. (46) holds at $s_+ = s_- = s$. This implies that the maximum of $m$ is achieved when no resampling/reweighting is applied.

It can be seen from the discussion so far that the symmetry is important in obtaining the above result. This explains the wide applicability of the result, and also inspires a further simplified model in the next subsection.

## 3.5 A further simplified model with more than two classes

Can the results so far be extended to the case with multiple classes more than two? While extending our problem setup to multiple classes and analyzing it using the replica method is possible, such an analysis would be rather complicated and we avoid doing it in this paper. Instead, in this subsection, we consider a further simplified model allowing us to analyze the multiclass problem easily.

From the discussion of the previous subsection, we understand that the symmetry in the loss and the problem setup is important to find the overlap maximum in the no-reweighting situation. On the basis of this understanding, reflecting the symmetry argument, we propose the following "one-dimensional" feature generation process for multiclass classification:

$$\boldsymbol{x}_\mu = t_\mu \frac{\boldsymbol{w}_0}{\sqrt{N}} + \boldsymbol{\xi}_\mu, \quad \mu = 1, 2, \ldots, M, \tag{54}$$

where the true feature is again normalized as $\|\boldsymbol{w}_0\|^2 = N$. The label $t_\mu \in \mathbb{R}$ is assumed to take one of $K$ values, where $K$ corresponds to the number of classes, and $\boldsymbol{\xi}_\mu$ is assumed to be an i.i.d. random variable following $\mathcal{N}(\boldsymbol{0}, \sigma^2 I_N)$ where $I_N$ is the identity matrix of dimension $N$. Here, only one true feature vector $\boldsymbol{w}_0$ governs the feature space, and thus the feature learning performance can be quantified by the accuracy of estimating $\boldsymbol{w}_0$ as in the case of binary classification discussed so far. For the sake of simplicity of the analysis, we consider, instead of a classification loss, a simple *unsupervised* loss for feature learning. Concretely, we consider $h_\mu = \boldsymbol{w}^\top \boldsymbol{x}_\mu$ and obtain the estimator by maximizing the variance of $\{h_\mu\}_{\mu=1}^M$ under the presence of a set of sample-wise reweighting factors $\{s_\mu\}_{\mu=1}^M$ satisfying $s_\mu \geq 0$ and $\sum_{\mu=1}^M s_\mu = 1$. The loss is formulated as

$$\mathcal{H}(\boldsymbol{w}) = -\sum_{\mu=1}^M s_\mu (h_\mu - \overline{h})^2 = -\boldsymbol{w}^\top A \boldsymbol{w}, \tag{55}$$

where $\overline{\cdot}$ denotes the weighted mean with the reweighting factors $\{s_\mu\}_{\mu=1}^M$ (i.e., $\overline{f} = \sum_{\mu=1}^M s_\mu f_\mu$), and where

$$A = \sum_{\mu=1}^M s_\mu (\boldsymbol{x}_\mu - \overline{\boldsymbol{x}})(\boldsymbol{x}_\mu - \overline{\boldsymbol{x}})^\top. \tag{56}$$

The minimizer of Eq. (55) under the normalization condition $\|\boldsymbol{w}\|^2 = N$ becomes our estimator of $\boldsymbol{w}_0$. Namely, the estimator is given by the eigenvector of the largest eigenvalue of $A$. The matrix $A$ can be rewritten as

$$A = \tau \boldsymbol{w}_0 (\boldsymbol{w}_0)^\top + \hat{\Sigma} + R, \tag{57}$$

where

$$\tau = \overline{t^2} - (\overline{t})^2, \tag{58}$$

$$\hat{\Sigma} = \overline{\boldsymbol{\xi}\boldsymbol{\xi}^\top} - \overline{\boldsymbol{\xi}}\,\overline{\boldsymbol{\xi}}^\top, \tag{59}$$

$$R = \left(\overline{t\boldsymbol{\xi}} - \overline{t}\,\overline{\boldsymbol{\xi}}\right)\boldsymbol{w}_0^\top + \boldsymbol{w}_0\left(\overline{t\boldsymbol{\xi}} - \overline{t}\,\overline{\boldsymbol{\xi}}\right)^\top. \tag{60}$$

The last term $R$ on the right-hand side of Eq. (57) is mean zero and is negligible if $M$ is large enough. Neglecting this term, we have our estimator as

$$\hat{\boldsymbol{w}} = \operatorname*{arg\,max}_{\boldsymbol{w}:\|\boldsymbol{w}\|^2=N} \left\{ \boldsymbol{w}^\top \left( N\tau \frac{\boldsymbol{w}_0}{\sqrt{N}} \left(\frac{\boldsymbol{w}_0}{\sqrt{N}}\right)^\top + \hat{\Sigma} \right) \boldsymbol{w} \right\}. \tag{61}$$

We furthermore assume that the reweighting factors $\{s_\mu\}_{\mu=1}^M$ are given depending on $M$ and satisfy $\lim_{M\to\infty} \sum_{\mu=1}^M s_\mu^2 = 0$. The empirical covariance $\hat{\Sigma}$ then converges to $\sigma^2 I$ as $M \to \infty$. Hence, in the large-$M$ limit, the maximizer $\hat{\boldsymbol{w}}$ approaches $\boldsymbol{w}_0$ irrespectively of the choice of $\{s_\mu\}_{\mu=1}^M$. Meanwhile, for large but finite $M$, the fluctuation of $\hat{\Sigma}$ becomes important. The fluctuation minimization leads to the best estimate of $\boldsymbol{w}_0$ and is achieved at the no resampling/reweighting situation $s_\mu = 1/M, \forall \mu$ on average. This can be easily shown by computing the mean deviation of the diagonal $\hat{\Sigma}_{ii}$ from $\sigma^2$, as

$$\mathbb{E}_{\boldsymbol{\xi}} \left(\hat{\Sigma}_{ii} - \sigma^2\right)^2 = \sigma^4 \left\{ 3\left(\sum_{\mu=1}^M s_\mu^2\right)^2 - 4\sum_{\mu=1}^M s_\mu^3 + 2\sum_{\mu=1}^M s_\mu^2 \right\}. \tag{62}$$

Minimization of the right-hand side under the constraint $\sum_{\mu=1}^M s_\mu = 1$ yields the uniform solution $s_\mu = 1/M, \forall \mu$, implying that no resampling/reweighing leads to the best feature learning performance. This provides a simple demonstration supporting Kang et al.'s observation in the multiclass case.

## 4 Numerical experiments

To verify the theoretical analysis using the replica method above, in this section we conduct numerical experiments and compare the result with the theoretical one. For this purpose, we only examined the case with CElo in this section since the CE loss is convex and thus the numerical optimization is relatively easy; such a good property is absent in the zero-one loss. The standard interior-point method was used for the optimization. In the following results, we conducted simulations with $N = 400$ and took a sample average over 100 different realizations of the dataset; the error bar is the standard error in the average. The parameter $\alpha$ was fixed to $\alpha = 20$, which is identical to the value used in sec. 3.3.

**Equivariance case**  We first experimented the equivariance case $\sigma_+ = \sigma_- = 0.6$. We start from plotting $m$ and $u$ against $b$ for $r_+ = 0.5, 0.2$ and $s_+ = 0.1, 0.5$ in Fig. 11. The agreement between the numerical results (markers) and the theoretical ones (lines) is excellent, justifying our theoretical treatment. Although our theoretical analysis assumes the high-dimensional limit $N \to \infty$, this numerical result indicates that several hundreds of $N$ can be regarded large enough.

For further quantification, we evaluated the loss-minimizing bias value and the corresponding $u_{\min}$ and $m(u_{\min})$ from the numerical experiments. Their plots, along with the theoretically-evaluated curves, are given in Fig. 12. The agreement between the numerical and theoretical results is again very good. Even the nontrivial location of the maximum point of $m(u_{\min})$ is reproduced by the numerical experiments.

The last result for the equivariance case is the plot at $b = 0$: $m(b = 0)$ and $u(b = 0)$ are plotted against $s_+$ with the theoretical curves in Fig. 13. The agreement is again good, and the maximum of $m(b = 0)$ is approximately obtained at the no resampling/reweighting case $s_+ = 1/2$, numerically supporting our main result in this paper.

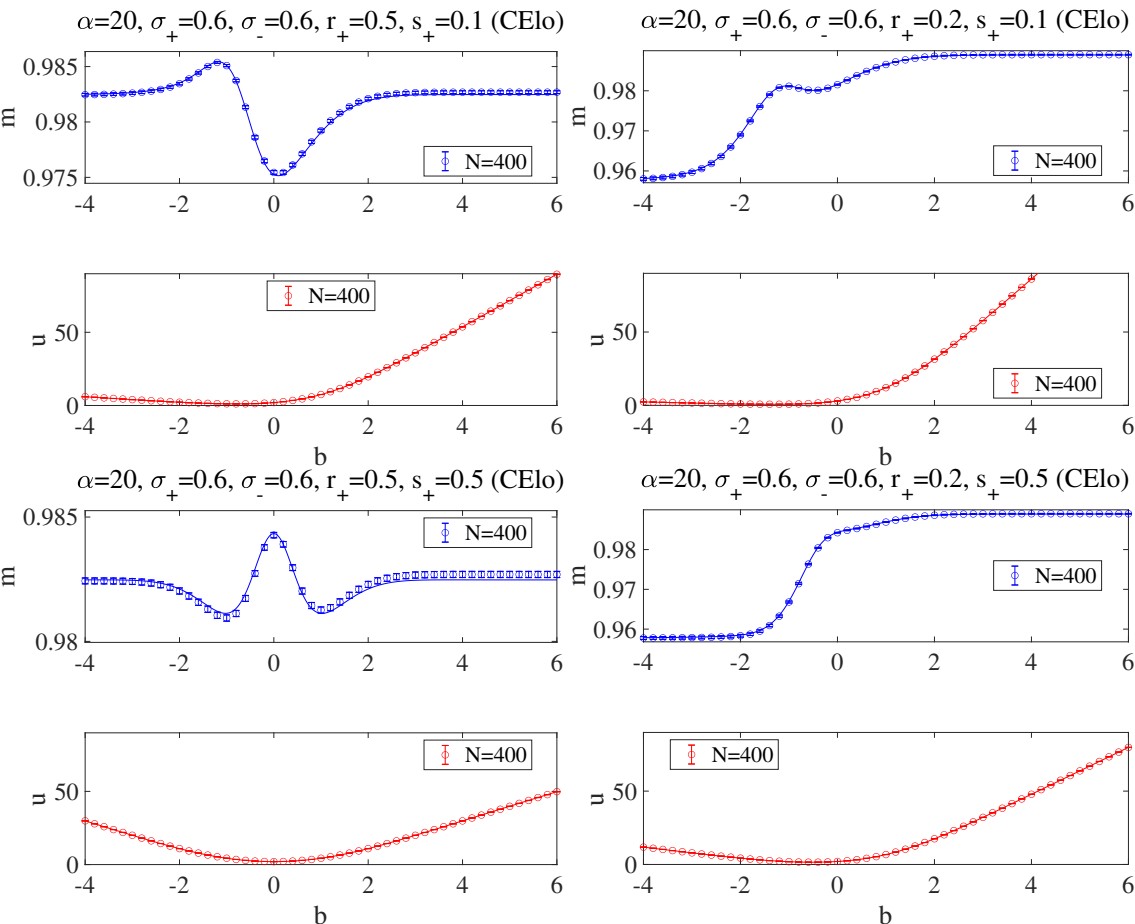

Figure 11: Plots of $m$ and $u$ against $b$ for $r_+ = 0.5$ (left) and 0.2 (right) and $s_+ = 0.1$ (upper) and 0.5 (lower) at the equivariance case $\sigma_+ = \sigma_- = 0.6$. The markers denote the numerical result and the lines represent the theoretical one. The agreement between them is fairly good.

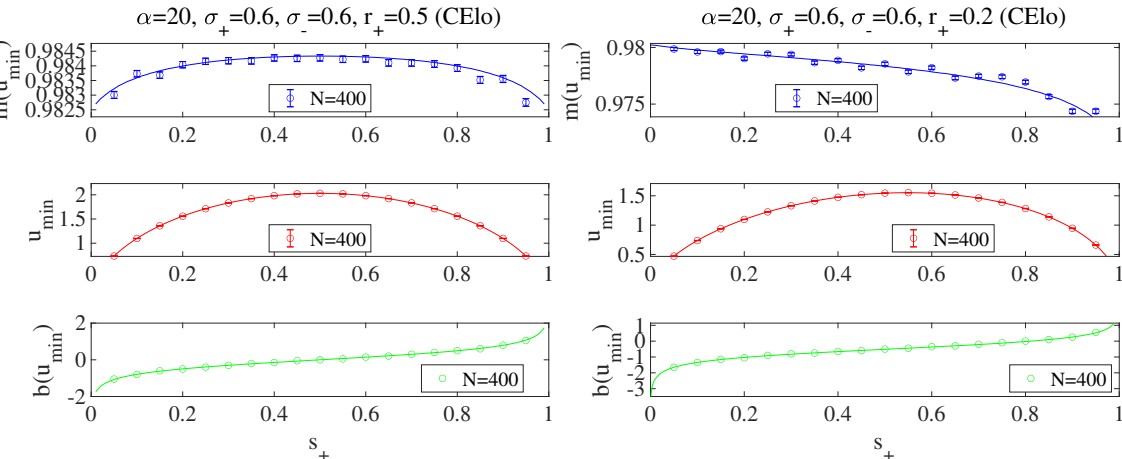

Figure 12: Plots of $m(u_{\min})$ and $u_{\min}$ against $s_+$ for $r_+ = 0.5$ (left) and 0.2 (right) at the equivariance case $\sigma_+ = \sigma_- = 0.6$. The agreement between the theoretical and numerical results is again very good.

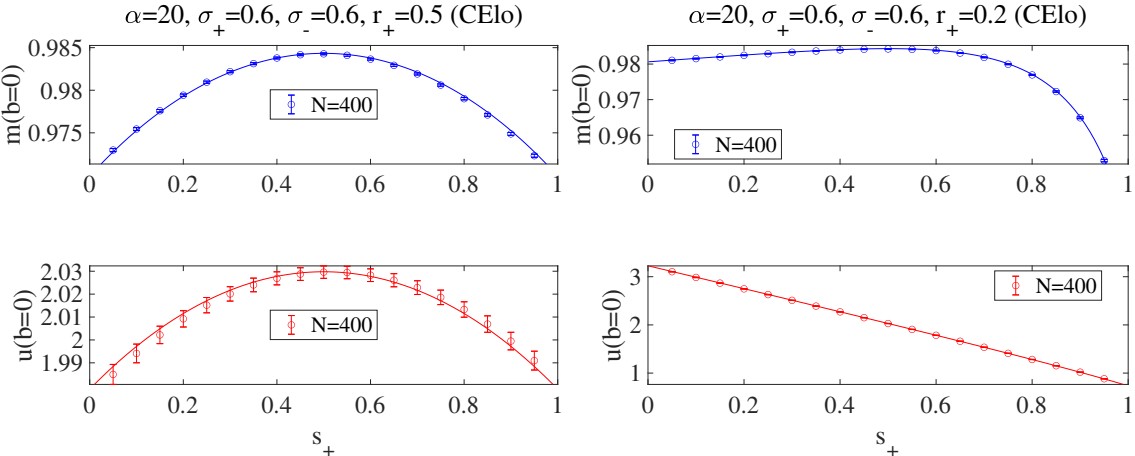

Figure 13: Plots of $m(b=0)$ and $u(b=0)$ against $s_+$ for $r_+ = 0.5$ (left) and $0.2$ (right) at the equivariance case $\sigma_+ = \sigma_- = 0.6$.

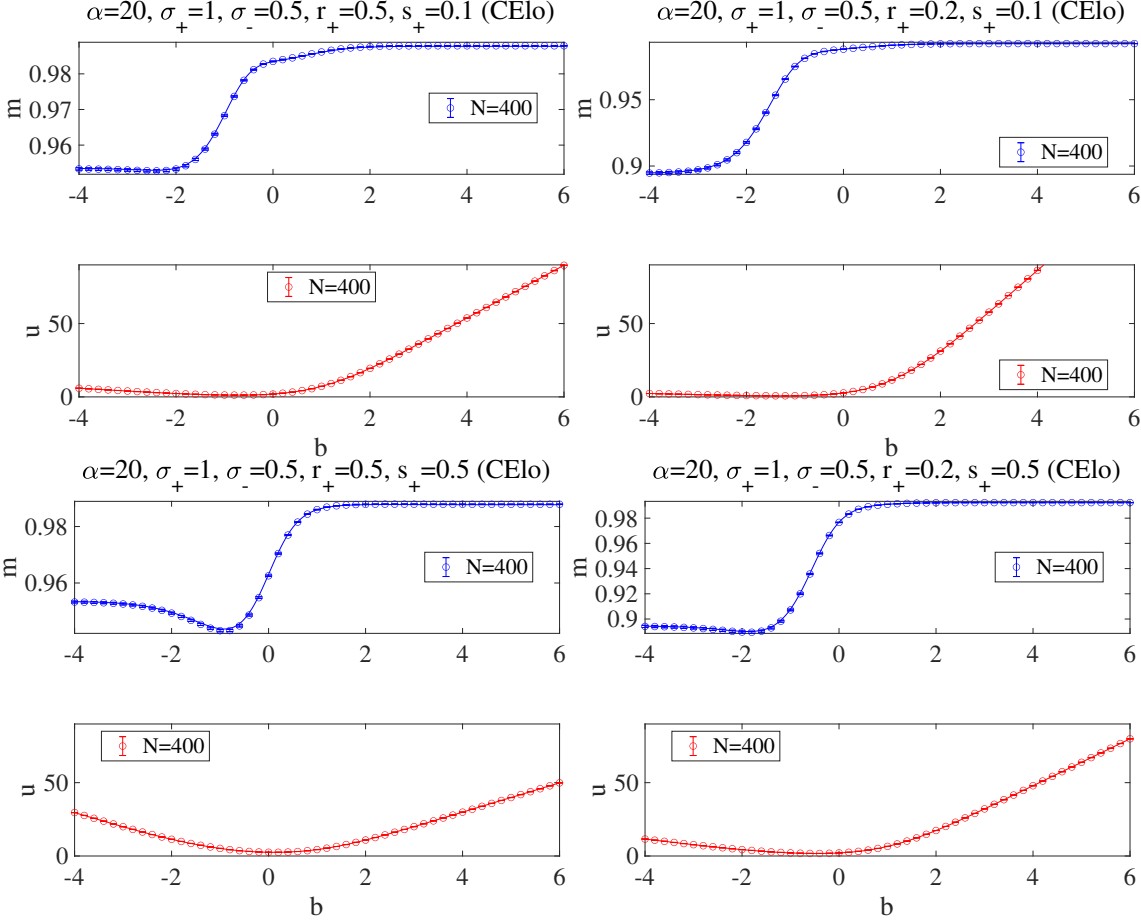

Figure 14: Plots of $m$ and $u$ against $b$ for $r_+ = 0.5$ (left) and $0.2$ (right) and $s_+ = 0.1$ (upper) and $0.5$ (lower) for the non-equivariance case $\sigma_+ = 1$, $\sigma_- = 0.5$.

**Non-equivariance case**   We next experimented the non-equivariance case $\sigma_+ = 1, \sigma_- = 0.5$, and show the results briefly. The plots of $m$ and $u$ against $b$ for $r_+ = 0.5$ and $0.2$ are shown in Fig. 14. Yet again, the agreement between the numerical and the theoretical results is fairly good. Our theoretical result is thus validated even for the non-equivariance case.

## 5   Conclusion

In this paper we have studied a toy model of binary classification for investigating the effect of the resampling/reweighting on the feature learning performance in the imbalanced classification; a special aim is at providing a theoretical basis for Kang et al.'s observation that the best performance of feature learning is achieved without any resampling/reweighting. The model's feature space is $\mathbb{R}^N$ and the class centers are assumed to be described by $\pm \boldsymbol{w}_0/\sqrt{N}(\in \mathbb{R}^N)$. The data generation is on the standard i.i.d. assumption with a label distribution having a control parameter of class imbalance and input distributions being allowed to have different variances on the two classes. In this setting, the feature learning performance can be quantified as the accuracy of estimating $\boldsymbol{w}_0$. The analysis of this and related quantities is conducted in the high-dimensional limit $N \to \infty$ keeping the dataset size ratio $\alpha = M/N$ finite, using the replica method from statistical mechanics.

Our theoretical analysis has revealed that the best performance of feature learning is actually achieved with no resampling/reweighting for a fairly wide range of losses and classifiers when input distributions are equivariance and the bias is set so that the decision boundary is located equidistantly from the two class centers: this is a desirable situation from the standard viewpoint of classification. The key of the derivation is the symmetry of the loss and the problem setup, explaining the wide applicability of the result. This provides a firm theoretical basis for Kang et al.'s observation and also implies that their learning result achieved the desirable situation. The emergence of such desirable situations as a result of learning may be understood from the viewpoint of NC (Papyan et al., 2020; Fang et al., 2021). More quantitative analysis has been conducted in two cases: the combinations of the cross-entropy loss and the logit classifier and of the zero-one loss and the perceptron. Although we have found that optimization of the energy or the overlap over the bias $b$ yields the better overlap than the above desirable situation, the selected bias values tend to take extreme values and also be highly dependent on the choice of loss, classifier, and the parameters. The practicality of such sensitive results should be tested by experiments in more realistic situations, which constitutes interesting future work.

Numerical simulations on the $N = 400$-dimensional systems have been also performed to check the validity of our theoretical results. The result showed good consistency with the theoretical one, which verifies our theoretical results and also reinforces the practicality of our theory derived in the limit $N \to \infty$.

As a future direction, it will be interesting to extend the analysis in this paper to multiple classes. Although we have proposed a further simplified model to treat multiclass classification on the basis of the insight from the analysis, the model is too simple in that it is essentially a one-dimensional problem with an unsupervised loss: this is incompatible with the standard treatment of multiclass classification. Such an extension would involve some difficulties related to the arrangement of the cluster centers. Some reasonable assumptions simplifying the analysis would be necessary, and the accumulated practical knowledge in machine learning communities would help find appropriate assumptions, just as Kang et al.'s observation which inspired the present study. Another interesting extension is to the semi-supervised setting with both labeled and unlabeled data. The effect of unlabeled data for performance is debatable from a theoretical viewpoint (Yang & Xu, 2020) while the benefit is evident in some applications (Jing & Tian, 2020). A toy-model study would be helpful to resolve this puzzle and to find conditions under which unlabeled data improve the performance.

## A  Volume computation

To handle the Dirac measures, we employ the following identities and trick:

$$1 = C_1 \int dQ \delta \left( NQ - \|\boldsymbol{w}_a\|^2 \right) = C_2 \int dQd\hat{Q} e^{\frac{1}{2}N\hat{Q}Q - \frac{1}{2}\hat{Q}\|\boldsymbol{w}_a\|^2}, \tag{63}$$

$$1 = C_1 \int dq \delta \left( Nq - \boldsymbol{w}_a^\top \boldsymbol{w}_b \right) = C_3 \int dqd\hat{q} e^{-N\hat{q}q + \hat{q}\boldsymbol{w}_a^\top \boldsymbol{w}_b}, \tag{64}$$

$$1 = C_1 \int dm \delta \left( Nm - \boldsymbol{w}_0^\top \boldsymbol{w}_a \right) = C_3 \int dmd\hat{m} e^{-N\hat{m}m + \hat{m}\boldsymbol{w}_0^\top \boldsymbol{w}_a}, \tag{65}$$

where the Fourier expression of the delta function is used to obtain the expression on the rightmost side of each equation, and where $C_1, C_2, C_3$ are normalization constants that are irrelevant in the following computations and will be discarded hereafter. Using these, we obtain

$$\begin{aligned}
V &= \int dQdqdmd\hat{Q}d\hat{q}d\hat{m} \; e^{N\left(\frac{1}{2}n\hat{Q}Q - \frac{1}{2}n(n-1)\hat{q}q - n\hat{m}m\right)} \\
&\times \mathrm{Tr}_{\{\boldsymbol{w}_a\}_{a=1}^n} e^{-\frac{1}{2}\hat{Q}\sum_{a=1}^n \|\boldsymbol{w}_a\|^2 + \hat{m}\sum_{a=1}^n \boldsymbol{w}_0^\top \boldsymbol{w}_a + \hat{q}\sum_{a<b} \boldsymbol{w}_a^\top \boldsymbol{w}_b} \\
&=: \int dQdqdmd\hat{Q}d\hat{q}d\hat{m} \; e^{N\left(\frac{1}{2}n\hat{Q}Q - \frac{1}{2}n(n-1)\hat{q}q - n\hat{m}m\right)} \mathcal{I}.
\end{aligned} \tag{66}$$

The factor $\mathcal{I}$ is computed as follows:

$$\begin{aligned}
\mathcal{I} &= \int \left( \prod_a d\boldsymbol{w}_a \delta(N - \|\boldsymbol{w}_a\|^2) \right) e^{-\frac{1}{2}\hat{Q}\sum_{a=1}^n \|\boldsymbol{w}_a\|^2 + \hat{m}\sum_{a=1}^n \boldsymbol{w}_0^\top \boldsymbol{w}_a + \hat{q}\sum_{a<b} \boldsymbol{w}_a^\top \boldsymbol{w}_b} \\
&= \int_{-i\infty}^{i\infty} \left( \prod_a \frac{id\Lambda_a}{2\pi} \right) e^{\frac{N}{2}\sum_a \Lambda_a} \\
&\times \int \left( \prod_a d\boldsymbol{w}_a \right) e^{-\frac{1}{2}\sum_{a=1}^n \Lambda_a \|\boldsymbol{w}_a\|^2 - \frac{1}{2}\hat{Q}\sum_{a=1}^n \|\boldsymbol{w}_a\|^2 + \hat{m}\sum_{a=1}^n \boldsymbol{w}_0^\top \boldsymbol{w}_a + \hat{q}\sum_{a<b} \boldsymbol{w}_a^\top \boldsymbol{w}_b} \\
&= \int_{-i\infty}^{i\infty} \left( \prod_a \frac{id\Lambda_a}{2\pi} \right) e^{N\left( \frac{1}{2}\sum_a \Lambda_a + \frac{1}{N}\sum_{i=1}^N \log \int Dz \prod_{a=1}^n \left( \int dw e^{-\frac{1}{2}\Lambda_a w^2 - \frac{1}{2}(\hat{Q}+\hat{q})w^2 + \hat{m}w_{0i}w + \sqrt{\hat{q}}zw} \right) \right)}.
\end{aligned} \tag{67}$$

Here, the integration w.r.t. $\Lambda_a$ can be replaced by its extremization condition thanks to the saddle-point/Laplace method. This yields the symmetric solution $\Lambda_a^* = \Lambda^*$. Hence,

$$\begin{aligned}
\mathcal{I} &= e^{N\left( \frac{1}{2}n\Lambda^* + \frac{1}{N}\sum_{i=1}^N \log \int Dz \left( \int dw e^{-\frac{1}{2}(\Lambda^* + \hat{Q} + \hat{q})w^2 + (\hat{m}w_{0i} + \sqrt{\hat{q}}z)w} \right)^n \right)} \\
&= e^{N\left( \frac{1}{2}n\Lambda^* + \frac{1}{N}\sum_{i=1}^N \log \int Dz \left( \sqrt{\frac{2\pi}{\Lambda^* + \hat{Q} + \hat{q}}} e^{\frac{1}{2}\frac{(\hat{m}w_{0i} + \sqrt{\hat{q}}z)^2}{\Lambda^* + \hat{Q} + \hat{q}}} \right)^n \right)}.
\end{aligned} \tag{68}$$

The last term in the exponent can be rewritten in the limit $n \to 0$ as

$$\begin{aligned}
&\lim_{n \to 0} \frac{1}{nN} \sum_{i=1}^N \log \int Dz \left( \sqrt{\frac{2\pi}{\Lambda^* + \hat{Q} + \hat{q}}} e^{\frac{1}{2}\frac{(\hat{m}w_{0i} + \sqrt{\hat{q}}z)^2}{\Lambda^* + \hat{Q} + \hat{q}}} \right)^n \\
&= \frac{1}{2}\log(2\pi) - \frac{1}{2}\log(\Lambda^* + \hat{Q} + \hat{q}) + \frac{1}{N}\sum_{i=1}^N \int Dz \frac{1}{2}\frac{(\hat{m}w_{0i} + \sqrt{\hat{q}}z)^2}{\Lambda^* + \hat{Q} + \hat{q}} \\
&= \frac{1}{2}\log(2\pi) - \frac{1}{2}\log(\Lambda^* + \hat{Q} + \hat{q}) + \frac{1}{2}\frac{\hat{m}^2 + \hat{q}}{\Lambda^* + \hat{Q} + \hat{q}},
\end{aligned} \tag{69}$$

where the relation $\sum_i w_{0i}^2 = N$ is used. Overall, the limit $n \to 0$ yields

$$
\lim_{n \to 0, N \to \infty} \frac{1}{nN} \log V(Q, q, m)
$$

$$
= \underset{\Lambda^*, \hat{Q}, \hat{q}, \hat{m}}{\mathrm{Extr}} \left\{ \frac{1}{2}\hat{Q}Q + \frac{1}{2}\hat{q}q - \hat{m}m + \frac{1}{2}\Lambda^* + \frac{1}{2}\log(2\pi) - \frac{1}{2}\log(\Lambda^* + \hat{Q} + \hat{q}) + \frac{1}{2}\frac{\hat{m}^2 + \hat{q}}{\Lambda^* + \hat{Q} + \hat{q}} \right\}. \tag{70}
$$

The extremization conditions w.r.t. $\hat{Q}$ and $\Lambda^*$ are degenerating. Rewriting $\hat{Q} \to \hat{Q} - \Lambda^*$ erases this degeneracy and makes the $\Lambda^*$ dependence very simple as $(-(1/2)Q + 1/2)\Lambda^*$ in the equation. The extremization condition w.r.t. $\Lambda^*$ thus yields $Q = 1$, leading to Eq. (27).

## B Interpretation of EOS

In this appendix, we provide an intuitive interpretation of the EOS (34) derived in sec. 3.2. We start with the first line of (67) for the factor $\mathcal{I}$. One can rewrite the exponent of the integrand as

$$
-\frac{1}{2}\hat{Q}\sum_{a=1}^n \|\boldsymbol{w}_a\|^2 + \hat{m}\sum_{a=1}^n \boldsymbol{w}_0^\top \boldsymbol{w}_a + \hat{q}\sum_{a<b} \boldsymbol{w}_a^\top \boldsymbol{w}_b = \frac{1}{2}\left\| \frac{\hat{m}}{\sqrt{\hat{q}}}\boldsymbol{w}_0 + \sqrt{\hat{q}}\sum_{a=1}^n \boldsymbol{w}_a \right\|^2 - \frac{\hat{m}^2}{2\hat{q}}\|\boldsymbol{w}_0\|^2 - \frac{\hat{Q}+\hat{q}}{2}\sum_{a=1}^n \|\boldsymbol{w}_a\|^2. \tag{71}
$$

Using the Hubbard-Stratonovich transform

$$
e^{c\|\boldsymbol{a}\|^2/2} = \left(\frac{c}{2\pi}\right)^{N/2} \int e^{-c\|\boldsymbol{z}\|^2/2 + c\boldsymbol{a}^\top \boldsymbol{z}} \, d\boldsymbol{z} \tag{72}
$$

with $c = \frac{1}{\hat{q}}$ and $\boldsymbol{a} = [(\hat{m}/\sqrt{\hat{q}})\boldsymbol{w}_0 + \sqrt{\hat{q}}\sum_{a=1}^n \boldsymbol{w}_a]/\sqrt{c}$, one can rewrite the integrand as

$$
e^{-\frac{\hat{Q}}{2}\sum_{a=1}^n \|\boldsymbol{w}_a\|^2 + \hat{m}\sum_{a=1}^n \boldsymbol{w}_0^\top \boldsymbol{w}_a + \hat{q}\sum_{a<b} \boldsymbol{w}_a^\top \boldsymbol{w}_b}
$$

$$
= e^{\frac{c}{2}\|\boldsymbol{a}\|^2 - \frac{\hat{m}^2}{2\hat{q}}\|\boldsymbol{w}_0\|^2 - \frac{\hat{Q}+\hat{q}}{2}\sum_{a=1}^n \|\boldsymbol{w}_a\|^2}
$$

$$
= \left(\frac{1}{2\pi\hat{q}}\right)^{N/2} \int e^{-\frac{1}{2\hat{q}}\|\boldsymbol{z}\|^2 - \frac{\hat{m}^2}{2\hat{q}}\|\boldsymbol{w}_0\|^2 + \hat{m}\boldsymbol{w}_0^\top \boldsymbol{z} + \sum_{a=1}^n \boldsymbol{w}_a^\top \boldsymbol{z} - \frac{\hat{Q}+\hat{q}}{2}\sum_{a=1}^n \|\boldsymbol{w}_a\|^2} \, d\boldsymbol{z}
$$

$$
= \left(\frac{1}{2\pi\hat{q}}\right)^{N/2} \int e^{-\frac{1}{2\hat{q}}\|\boldsymbol{z} - \hat{m}\boldsymbol{w}_0\|^2} \left( \prod_{a=1}^n e^{-\frac{1}{2\hat{q}}\|\boldsymbol{z} - \hat{q}\boldsymbol{w}_a\|^2 - \frac{\hat{Q}}{2}\|\boldsymbol{w}_a\|^2} \right) e^{\frac{n}{2\hat{q}}\|\boldsymbol{z}\|^2} \, d\boldsymbol{z}. \tag{73}
$$

This formula, after taking the limit $n \to 0$, implies that the problem of estimating $\boldsymbol{w}_0$ is equivalent, in the limit $N \to \infty$, to estimating it from its scaled and noisy version $\boldsymbol{z} = \hat{m}\boldsymbol{w}_0 + \boldsymbol{n}$ with Gaussian noise $\boldsymbol{n} \sim N(\boldsymbol{0}, \hat{q}I_N)$ by assuming the likelihood $\propto e^{-\frac{1}{2\hat{q}}\|\boldsymbol{z} - \hat{q}\boldsymbol{w}\|^2}$ and the prior $\propto e^{-\frac{\hat{Q}}{2}\|\boldsymbol{w}\|^2}$. The posterior distribution of $\boldsymbol{w}$ given $\boldsymbol{z}$ turns out to be $N\left(\frac{1}{\hat{Q}+\hat{q}}\boldsymbol{z}, \frac{1}{\hat{Q}+\hat{q}}I_N\right)$, and the posterior average with respect to this model turns out to be corresponding to the average $\langle(\cdots)\rangle$ over the Boltzmann distribution defined in Eq. (12).

Let $\hat{\boldsymbol{w}}, \hat{\boldsymbol{w}}'$ be independent samples from the posterior distribution $p(\boldsymbol{w} \mid \boldsymbol{z})$. One then has

$$
\frac{\mathbb{E}[\hat{\boldsymbol{w}}^\top \boldsymbol{w}_0]}{N} = \frac{1}{\hat{Q}+\hat{q}}\frac{\mathbb{E}[\boldsymbol{w}_0^\top \boldsymbol{z}]}{N} = \frac{\hat{m}}{\hat{Q}+\hat{q}}, \tag{74}
$$

$$
\frac{\mathbb{E}[\|\hat{\boldsymbol{w}}\|^2]}{N} = \frac{1}{(\hat{Q}+\hat{q})^2}\frac{\mathbb{E}[\|\boldsymbol{z}\|^2]}{N} + \frac{1}{\hat{Q}+\hat{q}} = \frac{\hat{Q}+2\hat{q}+\hat{m}^2}{(\hat{Q}+\hat{q})^2}, \tag{75}
$$

$$
\frac{\mathbb{E}[\hat{\boldsymbol{w}}^\top \hat{\boldsymbol{w}}']}{N} = \frac{1}{(\hat{Q}+\hat{q})^2}\frac{\mathbb{E}[\|\boldsymbol{z}\|^2]}{N} = \frac{\hat{m}^2 + \hat{q}}{(\hat{Q}+\hat{q})^2}, \tag{76}
$$

which should be equal to $m = [\langle \boldsymbol{w}\rangle^\top \boldsymbol{w}_0]_{D^M}/N$, $Q = [\langle \|\boldsymbol{w}\|^2\rangle]_{D^M}/N$, and $q = [\|\langle \boldsymbol{w}\rangle\|^2]_{D^M}/N$, respectively. These provide an interpretation of the EOS: In the limit $N \to \infty$, estimation of $\boldsymbol{w}$ may be regarded as being

performed with the Gaussian model defined as above, whose parameters $\hat{m}, \hat{Q}, \hat{q}$ are to be determined via a scalar estimation problem defined in terms of the loss $\ell$. The parameters $\hat{m}, \hat{Q}, \hat{q}$ should be taken so that the estimate of $\boldsymbol{w}$ has length $\sqrt{N}$, which implies that $Q = 1$ holds and hence $\hat{Q} + 2\hat{q} + \hat{m}^2 = (\hat{Q} + \hat{q})^2$. Under this condition one may forget the constraints on $\|\boldsymbol{w}_a\|^2$ in Eq. (67), as they will be satisfied automatically in the limit $N \to \infty$.

We have assumed the scaling of the order parameters in the limit $\beta \to \infty$ as in Eq. (29). In this limit, the signal-to-noise ratio of $\boldsymbol{z} = \hat{m}\boldsymbol{w}_0 + \boldsymbol{n}$ with $\boldsymbol{n} \sim N(\boldsymbol{0}, \hat{q}I_N)$ is $\hat{m}^2/\hat{q} = \tilde{m}^2/\tilde{\chi}$, which is finite, whereas the signal-to-noise ratio of $\boldsymbol{z}$ in the likelihood model $\propto e^{-\frac{1}{2\hat{q}}\|\boldsymbol{z}-\hat{q}\boldsymbol{w}\|^2}$ is $\hat{q} = \beta^2\tilde{\chi} \to \infty$, implying that the likelihood model is asymptotically noiseless. It can be understood as corresponding to the deterministic nature of the minimum-loss estimator $\hat{\boldsymbol{w}} = \arg\min_{\boldsymbol{w}:\|\boldsymbol{w}\|^2=N} \mathcal{H}(\boldsymbol{w} \mid D^M; b, s)$. One furthermore has

$$\hat{Q} + 2\hat{q} + \hat{m}^2 = (\hat{Q} + \hat{q})^2 \to \tilde{\chi} + \tilde{m}^2 = \tilde{Q}^2, \tag{77}$$

$$m = \frac{\hat{m}}{\hat{Q} + \hat{q}} \to m = \frac{\tilde{m}}{\tilde{Q}}, \tag{78}$$

$$q = 1 - \frac{1}{\hat{Q} + \hat{q}} \to q = 1 - \frac{1}{\beta\tilde{Q}}, \quad \chi = \frac{1}{\tilde{Q}}. \tag{79}$$

These reproduce Eqs. (34a)–(34c) of the EOS.

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
