# OpenReview forum: "When resampling/reweighting improves feature learning in imbalanced classification?: A toy-model study"
_TMLR — Rejected by TMLR_

### Review · Reviewer_Bax1 · 2024-09-15

**Summary Of Contributions:**

The paper concerns binary classification in high dimensions, focusing on the problem of resampling. By studying a series of fixed-point equations for a set of order parameters (derived by means of the replica method) the authors analyze the effect of resampling for different class balances and the relevance of the bias term in the estimation process. Moreover, they show that, under a series of suitable conditions, *no resampling* might be the optimal strategy, giving a theoretical foundation to recent findings of Cao et al (2019) and Kang et al (2019).

**Audience:**

Yes

**Broader Impact Concerns:**

I have no broader impact concerns as the contribution is mostly of a theoretical nature.

**Claims And Evidence:**

Yes

**Requested Changes:**

Aside from clarifying the points raised in the *Weaknesses* section above, I think that a number of contributions on the problem of classification appear to have been neglected by the authors and should be added to the manuscript for reference and comparison. The study of the classification of two clouds in the very same setup considered here (but without resampling) was studied by [Mignacco et al. (2020)](https://proceedings.mlr.press/v119/mignacco20a.html) and extended to the case of $K$ Gaussian clusters with generic, possibly different covariances by [Loureiro et al. (2021)](https://proceedings.neurips.cc/paper_files/paper/2021/hash/543e83748234f7cbab21aa0ade66565f-Abstract.html), and finally more recently to the case of fat-tailed clouds by [Adomaityte et al. (2023)](https://proceedings.neurips.cc/paper_files/paper/2023/hash/88be023075a5a3ff3dc3b5d26623fa22-Abstract-Conference.html). Incidentally, in the last contribution a dependence of the performance on the tail properties of the distribution is shown. Fairness models and the effect of unbalance have also been recently studied by [Sarao Mannelli et al. (2024)](https://openreview.net/forum?id=oupizzpMpY). All these contributions adopted a generic convex loss and solved the problem with a replica approach, so the equations considered by the authors in Section 3.2 should be essentially the same as those presented by [Mignacco et al. (2020)](https://proceedings.mlr.press/v119/mignacco20a.html), with a suitable ($s_y$-dependent) loss. From this point of view, I think the author should verify if their results are in agreement with these previous contributions and, if this is the case, state it explicitly.

In addition to this point, the hypothesis underlying the main result of the manuscript, presented in section 3.4, should be stated clearly and explicitly: e.g., what is the weakest assumption on the loss that guarantees the validity of the result? (Convexity, smoothness etc).

**Strengths And Weaknesses:**

**Strengths** The paper's main result is a theoretical justification for a recent observation of Kang et al. on the fact that classification tasks on unbalanced classes might not need resampling. The analysis is performed on a simple ``mean-field'' model which is analyzed by using the replica trick. All theoretical results are confirmed by numerical experiments.

**Weaknesses** The manuscript has, in my opinion, a series of weaknesses. I present below a list (please also refer to the *Requested changes* section).
- The set-up of the manuscript is quite well studied and its analysis via replicas has been performed in multiple papers of increasing generality (see the *Requested changes* paragraph). In light of this, I have the impression that the theoretical analysis in section 3, which is foundational for the main result of the manuscript, has little novelty. In this sense, the result in Section 3.4 itself might be seen as a minor (although possibly algebraically cumbersome) corollary of results available in the literature on such a classification task.
- In part of their analysis, the authors consider how $b$ and $u$ depend on $m_{\rm max}$ and $s_+$ for different relative cluster balances. However, it is not clear (at least to me) how this kind of analysis can give useful insights. As the authors themselves point out, in practical situations, it is not possible to directly search for the maximum overlap $m$ as the signal $\boldsymbol w_0$ is unknown, its estimate being the ultimate target... I have the impression therefore that this kind of analysis is confusing and not very informative, concerning the hypothetical situation in which some "informed" bias is plugged into the problem to direct us in the right direction. What instead usually occurs is that $b$ is estimated alongside $\boldsymbol w$ from $u$ by a minimization process: this second scenario is also considered in the manuscript alongside the aforementioned analysis and sounds to me more sensible. On top of this, I would have expected a study of the performances in terms of the prediction error of the estimator obtained for $u_{\rm min}$ as a function of $r$ by using $b(u_{\rm min})$ possibly as a function of $s$ to be kept as a hyperparameter.
- As the constraint on the weights is enforced in a hard way via a Dirac delta, the possible effect of a regularization (i.e., a soft constraint) is not obvious. I am pointing this out as the regularization is known to play a possibly beneficial role in the considered problem (see, eg. [Mignacco et al. (2020)](https://proceedings.mlr.press/v119/mignacco20a.html)).
- The authors choose to not investigate the dependence of the result on $\alpha$, which is perhaps a little bit unusual (see, again, the references given in the *Requested changes* section), in particular given the possible interpolation effects that might emerge for small $\alpha$. Also, standard quantities, such as the test error, of the estimator are not considered.
- As the authors acknowledge, the multiclass model presented in Section 3.5 is quite artificial and different from the setup considered in the paper. A natural extension would be instead the setup considered in a paper by  [Loureiro et al. (2021)](https://proceedings.neurips.cc/paper_files/paper/2021/hash/543e83748234f7cbab21aa0ade66565f-Abstract.html) where a genuine multiclass problem is studied. Nevertheless, I understand that the analysis in the latter setup can be cumbersome.

---

> ### Author Response · Authors · 2024-10-24
> **We would like to thank Reviewer Bax1 for his/her insightful comments and letting us know many useful references.**
>
> To Weaknesses:
>
> Below, we respond one-by-one to each of the pointed-out items, organized as an itemized list according to the order of appearance in the review comments.
>
> - We are aware that there are many papers analyzing similar problem settings using the replica method, but as long as we searched the literature there are no papers that discuss the accuracy of feature learning in the setup with class imbalance and the class-wise reweighted loss. Hence, the analysis of Sections 3 and 4 in our paper, which provides a replica-based proof of the empirical observation that the feature learning becomes optimal regardless of the degree of class imbalance when the no-resampling/reweighting is conducted, is novel. On the other hand, we completely agree that the procedure for deriving the free energy and the equations of state using the replica method is very well known and lacks technical novelty in this regard.
>
> - On this point, we have a clearly different opinion from the reviewer. While we understand that minimizing $u$ with respect to $b$ is practically what is being done, our consideration here suggests that this does not lead to the optimal feature learning. On the basis of this result, for instance, in the cases of class imbalance, it may be possible to achieve better feature learning by shifting the bias $b$ to an extreme value such as $b(m_{\rm max})$. As mentioned by the reviewer, knowing  $m_{\rm max}$ is not feasible in a realistic setting since we do not know $\boldsymbol{w}_0$. Even though it is the case, we can still try extreme values of $b$. It is difficult to examine such extreme values of $b$ solely via experimental trials/finding, because the cost of experiments is very expensive. In theory, however, we can explore various possibilities without much effort, and we think that even seemingly meaningless situations should be examined as well.
>
> - We would like to thank the reviewer for this comment. The reason why we constrain the norm of $\boldsymbol{w}$ in a hard manner is to quantify the feature learning with the overlap $m=\hat{\boldsymbol{w}}\cdot \boldsymbol{w}_0/N$. If the norm of $\hat{\boldsymbol{w}}$ is not constrained, then it becomes nontrivial how to quantify the feature learning performance. On the other hand, we know the nontrivial role of the regularizer in the respective problem and it is related to our problem setting. We are going to clarify how these two are related or different in the `Related work' section which will be introduced in the revised version.
>
> - This is because what we focus on is the quality of the feature learning, especially its dependence on the class imbalance and the reweighting factor. The change of $\alpha$ has some impact on the value of $m$ itself, but it has no effect on how $m$ depends on those parameters. The reason why the test error is not shown is because the test error of a classifier that is retrained on the basis of the acquired feature representation has a clear dependence on $m$, so that we think that it should be enough to show the dependence of $m$ on the parameters.
>
> - The reason why we do not consider the direct multiclass extension of the current setup in our paper is that, although it can be done, it is technically too cumbersome as the reviewer knows. That should require another long calculation as well as some additional consideration about how to concretize the problem setup details (such as the configuration of the multiple class centers). Hence we think that it should be deferred to another paper. We would kindly like to ask for your understanding on this point.
>
> For Requested Changes:
>
> We would thank the reviewer for letting us know many useful earlier works which were not cited in the present manuscript. We are going to introduce a new subsection of `Related work' and explain the importance of the suggested references as well as the difference from our present work. For Sections 3 and 4, we will clearly state the assumptions necessary for the result (actually we do not need either convexity or smoothness for the loss).

---

> > ### Comment · Reviewer_Bax1 · 2024-10-25
> >
> > I would like to thank the authors for their detailed reply. Concerning the originality of the contribution, the main difference I see in the treatment of the problem with respect to previous works is, more than the presence of class imbalance, the introduction of the parameters $s_\pm$ which add an effectively single additional reweighting factor to be learned. I understand that how much this (low dimensional) additional complication adds to the previous technical results is perhaps a matter of personal taste: nevertheless, I recommend the authors to more clearly refer to previous works that considered the same setup using the very same technical approach up to minimal differences. With respect to the exploration of extreme values of $b$, I still have the impression that part of the discussion, e.g., the analysis of $b(m_{\rm max})$, might give effectively very minimal insights as it is not clear if such regimes are achievable, or how.
> >
> > I have read the reports of the other reviewers and I also agree with Reviewer MA55 about the required improvement of the text exposition. I changed my *Audience* evaluation from *No* to *Yes* as my former opinion was mostly based on the technical part.
> >
> > Finally, with respect to the last point, I agree with the authors that the multiclass case can be deferred to future works, as mine was more a comment than a request.

---

> > > ### Author Response · Authors · 2024-10-28
> > >
> > > Thank you for your detailed feedback and constructive comments. We will clarify how our approach is different from existing literature, which uses similar setups and methods. We will also improve the exposition of the text to make the presentation more accessible to a broader audience. Regarding the exploration of extreme values, we recognize the importance of addressing realistic and interpretable parameter regimes. We will clarify this point as much as possible in the revision.
> > >
> > > Thank you again for your thoughtful review and recommendations, which help us enhance the clarity and relevance of our paper.

---

### Review · Reviewer_MA55 · 2024-10-11

**Summary Of Contributions:**

The authors consider a simple model of binary classification to characterize the effect of resampling or reweighting when one of the two classes is underrepresented. The authors' analysis takes inspiration from previous observations, e.g. [Kang et al. 2019], that show empirically how resampling is not effective in enhancing the model feature learning capabilities . They consider the asymptotic properties of Empirical Risk Minimizers $\hat{w}$ through the replica method coming from statistical physics, and they provide a theoretical explanation for the findings of [Kang et al. 2019] by looking at the scalar product $w_0^\top \hat{w}$, with $w_0$ the ground truth. The authors also consider cases in which resampling is actually effective, i.e., when the two classes do not share the same variance. The authors present numerical simulations supporting the theoretical claims.

**Audience:**

Yes

**Broader Impact Concerns:**

I do not have any concern on this submission as its interest is primarily theoretical.

**Claims And Evidence:**

Yes

**Requested Changes:**

The paper is far from being ready for publication and the amount of changes needed brings me to advocate for rejection of the present submission.

## Major points

- First, I would like to acknowledge that I correctly followed the theoretical computation based on the replica method, and the authors do a great job in explaining different highly technical details.
However, the extensive focus on the computation in the main text (pages 5-8) makes impossible for the non-expert reader to grasp the key contribution of this work. While the theoretical approach through the replica method is nice, this should not occupy the core of the main text. The authors, starting from page 8 onwards, refer to quantities introduced in a previous highly technical computation that completely penalize the readability of the manuscript, e.g. in page 8, computing $v_{\pm}$; at this point the reader who is not interested in the technical details is forced to stop following.

- The main contribution of this paper are in my opinion: a) providing a theoretical explanation for the findings of  [Kang et al. '19]; b) showing that such findings are restricted to the equivariance case. This fact is not evident; while point (a) is mentioned in the introduction, point (b) is thoroughly discussed only in Page 17.

- The authors do not provide correct referencing to the related literature. While different pointers are provided below, my main concern is how the present submission compare with a similar work dealing with statistical physics approach to classification in unbalanced setting [1].

        [1] Loffredo et al., Restoring balance: principled under/oversampling of data for optimal classification.

- Moreover, what is the technical novelty of the replica computation over [2]? Given that the authors reserve a core space to the computation I would expect to be novel but I fail to grasp to which extent this is true.

         [2] Loureiro et al., Learning Gaussian Mixtures with Generalised Linear Models: Precise Asymptotics in High-dimensions.

- [Barkai & Sompolinky, 1994] is the only reference provided for theoretical treatments of binary classification. This needs certainly to be included in a broader context.

- Many interesting references are missing on the dynamics under class imbalance, e.g. [3], and Bayes optimal approaches or spectral approaches, e.g., see [4,5].

        [3] Francazi et al., A Theoretical Analysis of the Learning Dynamics under Class Imbalance.
        [4] Lesieur et al., Constrained low-rank matrix estimation: phase transitions, approximate message passing and applications.
        [5] Jinho Baik et al.. Phase transition of the largest eigenvalue for nonnull complex sample covariance matrices.


## Minor points

- Page 1: Remove punctuation after question mark in the title.
- Page 1: define representation and feature learning capabilities.
- Page 2: "irrespectively of the degree of the class imbalance"; is this formally correct? Should the authors specify the scaling of the class imbalance with $N$?
- Page 3: Weak justification for eq. (4). One must have stronger argument than simply recalling CLT.
- The current status of the caption is not detailed. The reader sees many plots without understanding which ones are really important for the contirbution of the paper. I would suggest to move many of the plots in the appendix.

**Strengths And Weaknesses:**

The main strength of the submission is the interesting motivation behind the theoretical analysis. Understanding the effects of resampling/reweighting from a theoretical perspective is certainly welcome.

The strong weakness is the clarity of the exposition and the framing of the present submission in terms of the related literature. The structure of the main text needs to be revisited completely and the exposition requires several changes in different parts. More details are provided below.

---

> ### Author Response · Authors · 2024-10-24
> **We would like to thank Reviewer MA55 for pointing out unclear points and letting us know many useful references.**
>
> To Weaknesses:
>
> We are sorry for our unclear expositions. The detailed calculations in the main text certainly makes it difficult to read, especially for those unfamiliar with the technicalities. However, since the ``proof'' in Section 3.4 is one of the main outcomes of the paper, we think it inappropriate to move all the technical contents to the Appendix. Hence, in the revision, we will do our best to break the current structure into an outline and details, and keep only the outline in the main text and move the details to the Appendix.
>
> To Requested Changes:
>
> Below, we respond one-by-one to each of the pointed-out items, organized as an itemized list according to the order of appearance in the comments.
>
> # Major
> - We would like to thank the reviewer for his/her feedback. As mentioned above, we will improve readability by separating the key parts from the technical details as much as possible, and moving the latter to the Appendix.
>
> - In the revision, to make point b) clearer, we will clearly state the conditions for point a) to hold in the introduction.
>
> - We would like to thank the reviewer for letting us know the important reference. We will cite [1] and clarify its relation with and difference from our present result in the newly added `Related work' section in the revision.
>
> - Our problem setup considers both the class imbalance and the class-wise loss reweighting, and their effect on the feature learning performance is analyzed. This setup and the derived consequences are definitely novel, but there is no newly developed technique to cover this setup, and hence the technical novelty in the replica computation over [2] is tiny. We will make this point clear in the `Related work' section.
>
> - We will explain the contribution of [Barkai & Sompolinky, 1994] in more details in the `Related work' section.
>
> - We will also cite [3-5] and will clarify the positioning of those papers in comparison to this paper in the `Related work' section.
>
> # Minor
> - Thank you for pointing this out. We will remove the punctuation accordingly.
>
> - We are not sure if we understand your point correctly. Our feature is defined to be $h=\boldsymbol{w}^{\top} \boldsymbol{x}$ as explained in Section 2, while we do not perform any representation learning: Our input vector $\boldsymbol{x}$ should be interpreted as a last-layer representation when comparing our model with the common experimental setup using deep neural networks. What we did is to clarify that when the representation distribution is equivariance, the feature learning becomes the best with the no resampling/reweighting situation. In the revision, we will improve the main text to make this point clearer.
>
> - Class imbalance is controlled by $r_+$ and our main result of the best feature learning with no resampling/reweighting situation is correct irrespectively of the value of $r_+$. No scaling w.r.t. $N$ is necessary.
>
> - We are sorry again if we miss your point. Equation (4) is derived directly from our Gaussianity assumption of $\boldsymbol{\xi}$, and no justification is considered to be necessary. The many plots are given to provide a thorough qualitative comparison of different situations, to clarify the complicated interplay among loss, classifier, and reweighting factor. This complicacy prevents us from having a clear interpretation/description about the role of reweighting factor in general situations, except for the case with the equivariance representation distribution and the decision plane equidistant from the class centers. We think that the clarification of the lack of simple interpretation in general situations is one of the present results, and think these many plots are necessary to support this non-simple conclusion.

---

> > ### Comment · Reviewer_MA55 · 2024-10-25
> >
> > I thank the authors for their detailed response.
> >
> > I support their decision of not moving all the technical details in the appendix, but the main text should be readable also for the non-expert reader.
> >
> > Regarding the [Barkai & Sompolinski, 1994], I am not suggesting explaining that contribution in detail but include the numerous works on statistical physics of learning that have been published from the '90s up to contemporary days (see e.g. reference [1] and the many references pointed out by Reviewer Bax1).

---

> > > ### Author Response · Authors · 2024-10-28
> > >
> > > Thank you for your valuable feedback and supporting our approach to presenting technical details.
> > >
> > > Regarding the inclusion of references on the statistical physics of learning, we will incorporate additional references, particularly from foundational works up to recent contributions in the `Related work' section in the revision. This will help highlight the progression of ideas and the relevance of our approach in the field.
> > >
> > > Thank you again for your insightful comments and recommendations.

---

### Review · Reviewer_EkgP · 2024-10-14

**Summary Of Contributions:**

The motivation of this paper is to develop a toy binary classification model that attempts to explain an interesting observation discussed in papers such as *Cao et al. (2019)*; *Kang et al. (2019)*, etc. The observation in question is that the best features are learned *without* any kind of re-sampling/re-weighting in unbalanced binary classification problems, which is both surprising and counter-intuitive. To explain this odd phenomenon, the authors propose a toy binary classification model in which they theoretically prove that the no re-sampling/no re-weighting condition yields the best features subject to a set of assumptions.
The assumptions are as follows:
1. The positive and negative class-centers are located at $\frac{+w_0}{\sqrt(N)}$ and $\frac{-w_0}{\sqrt(N)}$ respectively ($N$=number of dimensions)
2. The number of dimensions $N \rightarrow ∞$ and the ratio $\frac{M}{N}$ is finite ($M$=number of samples)
3. The loss must be symmetric i.e. $l(+h,+y)=l(-h,-y)$ where $h$ is the hypothesis function of the form $\frac{w^T \cdot x}{\sqrt{N}} + b$
4. The positive and negative classes have the same variance $\sigma_{+}^2 = \sigma_{-}^2$
5. The bias is set such that the decision boundary is equidistant from both class centers

The authors then proceed to lay out their framework in great detail and provide a formal proof of their result using the replica method. In addition, the authors also provide a detailed discussion of various configurations of their model including the one where the result of interest is observed. Finally, the authors run some numerical experiments and show the agreement between their experiments and their theoretical results for $N=400$.

**Audience:**

Yes

**Claims And Evidence:**

Yes

**Requested Changes:**

In addition to the weaknesses above, I would like to see the following points clarified as well:
1. I am not sure I understand how Equation (6) is equivalent to the sigmoid function. Should it be $\frac{e^{h/2}}{2cosh(h/2)}$? Could the authors clarify this?
2. For those coming from a non-statistical mechanics background, it would probably be a good idea to explain why a high value of the overlap parameter $m$ corresponds to a good learned representation. Although a full-blown expression would exceed the scope of the paper, a short paragraph or two might be helpful to make the results more accessible to a general audience.
3. Although again probably out of the scope of the paper, I think the readers might be interested to hear about the authors' opinions of how their results would hold (or break) if the clusters centers were $w_{0+}$ and $w_{0-}$ where $w_{0+} \ne -w_{0-}$. If the class variances are still equal and the biases are chosen appropriately so that the decision boundary is equidistant, do the authors conjecture that their result would still hold? This change/clarification isn't necessary for the acceptance of the paper, but it might be a nice discussion point to have.

**Strengths And Weaknesses:**

**Strengths:**
1. The paper is very thorough and well-written. The authors spend a considerable amount of time explaining how their framework is structured and gradually build up to the proof. Section 3, in particular, walks the readers through various possible configurations of the model and contains many interesting observations including the observation of interest.
2. The limitations of the model as well as practical considerations that need to be addressed are also touched upon. This is very important, since although the scope of the paper is limited to the study the theoretical condition(s) under which no re-sampling/no re-weighting configuration yield the best features, having a clear understanding of the assumptions/limitations may help authors of future work to determine (or at least approximate) how many of these assumptions are met in real-world data.
3. The authors also run a set of numerical experiments to provide the reader with some empirical evidence for their claims when the assumptions of the model are met.

**Weaknesses:**
Most of the weaknesses I will cover here are questions that I need further clarification from the authors. More specifically:
1. To my understanding, the authors' toy model is formulated as $M(yh)$ where $h$ is a linear mapping of $x$ and $M$ is a symmetric activation function. The formulation then provides the basis for the proof that is then used to derive the result. The no-resampling/no-reweighting experiments in Cao et al. uses bidirectional LSTMs while Kang et al. uses variants of ResNet. I am skeptical as to whether the observations in these experiments can be attributed to the condition discussed in this paper. Could the authors clarify this? Also, it might be a good idea to clarify that the authors' results should apply to GLMs in general.
2. From the main proof in Section 3.4, it would seem that the no re-sampling/no re-weighting condition should hold regardless of the value of $\alpha$ (as long as it is finite). Could the authors clarify further why they chose $\alpha=20$ for both the theoretical analysis as well as the numerical experiments? Also, what were the cluster centers chosen (how close are far apart were they)?
3. The numerical results are a nice addition to provide some sort of empirical evidence that the model configuration used for the theoretical results in Section 3 hold for $N=400$. I currently interpret this section as a set of results for a particular configuration of the model where $N=400$. It could be the case that the agreement would get worse with other configurations even when $N=400$ so it is difficult to draw any general conclusions about how high N needs to be in order for the no-resampling/no-reweighting condition to apply. Is the purpose of this section to just do an apples-to-apples comparison with the same model configuration used in Section 3? If so, then it might be a good idea to clarify this at the beginning of Section 4.

---

> ### Author Response · Authors · 2024-10-24
> **We would like to thank Review EkgP for his/her positive comments and clarification of misleading points.**
>
> To Weaknesses:
> 1. First of all, please note that in the experiments of Cao et al., as well as those of Kang et al., whatever backbone networks (bidirectional LSTM, variant of ResNet, etc.) were used as feature extractors, the final layer of their models are linear classifiers. We therefore think that the issue raised by the reviewer is related to whether the input vector $x$ in our model corresponds, at the distribution level, to the feature $h$ in the final layer of their models. In other words, if the distribution of $h$ has equal variance across classes, the situation is ideal and can be compared with our result. Since the class-wise variances of $h$ have not been shown in their papers, we do not know how they actually were, but this point can be verified in principle. Moreover, it is known that in DNNs trained with a large amount of training samples and an enough amount of time, the features in the final layer converge to a single point for each class. This phenomenon is called Neural Collapse (NC) (Papyan et al., 2020). This suggests that the assumption of equal variance is satisfied in the limit where the variance is zero. Although this point is already mentioned in the main text, we will revise the manuscript to make it clearer.
>
> 2. There is no particular reason why we chose $\alpha=20$. As mentioned at the beginning of Section 3.1.1, as far as we have experimented, any other values will be fine as long as they are not too small. We placed the cluster centers $\pm\boldsymbol{w}_0/\sqrt{N}$ symmetrically about the origin, in view of Neural Collapse (see item 1 above). How they are far apart is a matter of scaling: We fixed the squared norm of $\boldsymbol{w}_0$ to be $N$, and set the standard deviation $\sigma$ to values with which the class-conditional input distributions exhibit reasonable level of overlap.
>
> 3. The purpose of numerical experiments in Section 4 is to verify whether the theoretical analysis performed in Section 3 is correct. Our theoretical analysis is based on a method that is valid in the limit $N \to \infty$, so that if the numerical experiments with sufficiently large $N$ are consistent with those theoretical results, it would support the correctness of the theoretical analysis. In fact, the results were consistent. The parameter settings are basically the same as in Section 3 and hence it is the same model configuration. We will clarify this point at the beginning of the section.
>
> To Requested Changes:
> 1. We would like to thank the reviewer for this comment. The "logit" function is usually defined as $\log\frac{p}{1-p}$, which is the *inverse* of the logistic function, so that our "logit" function was a misnomer. We will change its name in the revised manuscript. The function $\frac{e^{h/2}}{2\cosh(h/2)}$ mentioned by the reviewer is exactly the same as our equation (6) by appropriately rescaling $h$, so that there is no need to distinguish them.
>
> 2. Since we fix the squared norm of $\boldsymbol{w}$, the overlap parameter quantifies how well $\boldsymbol{w}$ aligns with $\boldsymbol{w}_0$, the latter of which is the most discriminative direction in our setting. We will point it out in the revised manuscript.
>
> 3. Even in such a case, by setting the midpoint of $w_{0+}$ and $w_{0-}$ at the origin, we would get the same result as the current one. However, if we do not reset the midpoint and perform the same analysis as the current one with fixing the norm of $\boldsymbol{w}$, the result will be different. But we think this case has no significant meaning. This is because our feature $\boldsymbol{h} = \boldsymbol{w}^{\top}\boldsymbol{x}$ is a projection of the input $\boldsymbol{x}$ onto a specific vector, and only the projection onto the line connecting $w_{0+}$ and $w_{0-}$ contributes to classification since the distribution of our input vector distribution given the label is spherically symmetric. The information in the orthogonal subspace to this line is of no use for classification. In fact, if we analyze the same problem with a $\ell_2$ regularization to the $\boldsymbol{w}$ norm instead of the current hard constraint, the orthogonal subspace would be suppressed by the regularization and only the one-dimensional space connecting $w_{0+}$ and $w_{0-}$ would be left as meaningful.

---

> > ### Comment · Reviewer_EkgP · 2024-10-25
> >
> > I would like to thank the authors for their detailed response.
> > My questions/concerns have been sufficiently addressed and I am ready to make a decision.

---

> > > ### Author Response · Authors · 2024-10-28
> > >
> > > Thank you very much for your thoughtful review and for considering our responses. We appreciate your time and effort in evaluating our work.

---

### Author Response · Authors · 2024-10-24
**We thank the reviewers for all the constructive comments.**

First of all, we would like to thank all the reviewers for their review comments. For those major points that most of the reviewers are concerned about, when an opportunity of revision is given to us, we will make the following corrections to improve the manuscript:
1. Adding sentences clarifying the relationship between the results of Cao/Kang et al.’s papers and our results.
2. Adding an explanation of why  $m = \frac{w_0 \cdot w}{N}$  is important.
3. Moving the detailed calculations in Section 3 to the Appendix, leaving only the outline in the main text.
4. Emphasizing that the equivariance condition is required for the optimal feature learning to occur in Introduction.
5. Adding the references that were pointed out by the reviewers, and also adding a Related Work section to summarize the results of those papers and explaining how they differ from the results of this paper.
6. Clarifying the purpose of the numerical experiments in Section 4 at the beginning.
7. Removing the punctuation in the title.

---

### Decision · Action_Editor_idcR · 2024-11-21

**Recommendation:** Reject

**Comment:**

While the relevance and theoretical analysis were appreciated by the reviewers, significant concerns were raised regarding the clarity, accessibility, and contextualisation of this work. For this reason, I believe this manuscript could benefit from a major rewritting, and my recommendation is therefore towards rejection and resubmission.

In particular, it could benefit from the following constructive suggestions from the reviewers:

- Reorganizing the text to prioritize key insights and improve accessibility. For example, as Reviewer MA55 notes, the extensive focus on the replica computation (Section 3) penalizes readability, and moving technical details to an appendix would enhance clarity.

- A clearer comparison with related works as highlighted by Reviewers MA55 and Bax1. Explicitly state the novelty of your findings and how they differ from these prior studies.

- A clarification of the assumptions underlying your main results, such as the role of the loss function's properties (e.g., convexity, smoothness).

- Refining the exposition of key findings, such as explicitly discussing the implications of your results for real-world scenarios, as Reviewer EkgP suggests, and clarifying the connection between numerical experiments and theoretical predictions.

**Audience:**

The paper is in a topic of interest to the audience of TMLR interested in high-dimensional statistics and machine learning theory.

**Claims And Evidence:**

The claims made in the submission are partially supported by accurate and convincing evidence, but there are some limitations in clarity and depth. The theoretical analysis is solid and corroborated by numerical experiments. However, some reviews pointed the manuscript lacks sufficient contextualization and comparison with closely related works.

**Resubmission Of Major Revision:**

The authors may consider submitting a major revision at a later time.